# Sample-Efficient Reinforcement Learning Is Feasible for Linearly Realizable MDPs with Limited Revisiting

**Gen Li**
Princeton

**Yuxin Chen**
Princeton

**Yuejie Chi**
CMU

**Yuantao Gu**
Tsinghua

**Yuting Wei**
UPenn

## Abstract

Low-complexity models such as linear function representation play a pivotal role in enabling sample-efficient reinforcement learning (RL). The current paper pertains to a scenario with value-based linear representation, which postulates linear realizability of the optimal Q-function (also called the "linear $Q^\star$ problem"). While linear realizability alone does not allow for sample-efficient solutions in general, the presence of a large sub-optimality gap is a potential game changer, depending on the sampling mechanism in use. Informally, sample efficiency is achievable with a large sub-optimality gap when a generative model is available, but is unfortunately infeasible when we turn to standard online RL settings.

We make progress towards understanding this linear $Q^\star$ problem by investigating a new sampling protocol, which draws samples in an online/exploratory fashion but allows one to backtrack and revisit previous states. This protocol is more flexible than the standard online RL setting, while being practically relevant and far more restrictive than the generative model. We develop an algorithm tailored to this setting, achieving a sample complexity that scales polynomially with the feature dimension, the horizon, and the inverse sub-optimality gap, but not the size of the state/action space. Our findings underscore the fundamental interplay between sampling protocols and low-complexity function representation in RL.

## 1   Introduction

Emerging reinforcement learning (RL) applications necessitate the design of sample-efficient solutions in order to accommodate the explosive growth of problem dimensionality. Given that the state/action space could be unprecedentedly enormous, it is often infeasible to request a sample size exceeding the fundamental limit set forth by the ambient dimension in the tabular setting (which enumerates all combinations of state-action pairs). As a result, the quest for sample efficiency cannot be achieved in general without exploiting proper low-complexity structures underlying the problem of interest.

### 1.1   Linear function approximation

Among the studies of low-complexity models for RL, linear function approximation has attracted a flurry of recent activity, mainly due to the promise of dramatic dimension reduction in conjunction with its mathematical tractability (see, e.g., Wen and Van Roy (2017); Yang and Wang (2019); Jin et al. (2020); Du et al. (2020a)). Two families of linear function approximation merit particular attention, which we single out below. Here and throughout, we concentrate on a finite-horizon Markov decision process (MDP), and denote by $\mathcal{S}$, $\mathcal{A}$ and $H$ its state space, action space, and horizon, respectively.

- *Model-based linear representation.* Yang and Wang (2019); Jin et al. (2020); Yang and Wang (2020) studied a scenario when both the probability transition kernel and reward function of the MDP can be linearly parameterized. This type of MDPs is commonly referred to as linear MDPs. Letting $P_h(\cdot \mid s, a)$ be the transition probability from state $s$ when action $a$ is executed at step $h$,

35th Conference on Neural Information Processing Systems (NeurIPS 2021).

the linear MDP model postulates the existence of a set of predetermined $d$-dimensional feature vectors $\{\varphi_h(s,a) \in \mathbb{R}^d\}$ and a set of unknown parameter matrices $\{\mu_h \in \mathbb{R}^{d \times |\mathcal{S}|}\}$ such that

$$\forall (s,a) \in \mathcal{S} \times \mathcal{A} \text{ and } 1 \leq h \leq H: \qquad P_h(\cdot \,|\, s,a) = \big(\varphi_h(s,a)\big)^\top \mu_h. \qquad (1)$$

Similar assumptions are imposed on the reward function too. In words, linear MDP posits that the probability transition matrix is low-rank (with rank at most $d$) with the column space known *a priori*, which forms the basis for sample size saving compared to the unstructured setting.

- *Value-based linear realizability.* Rather than relying on linear embedding of the model specification (namely, the transition kernel and reward function), another class of linear representation assumes that the action-value function (or Q-function) can be well predicted by linear combinations of known feature vectors $\{\varphi_h(s,a) \in \mathbb{R}^d\}$. A concrete framework of this kind assumes linear realizability of the optimal Q-function (denoted by $Q_h^\star$ at time step $h$ from now on), that is, there exist some unknown vectors $\{\theta_h^\star \in \mathbb{R}^d\}$ such that

$$Q_h^\star(s,a) = \big\langle \varphi_h(s,a), \theta_h^\star \big\rangle \qquad (2)$$

holds for any state-action pair $(s,a)$ and any step $h$. It is self-evident that this framework seeks to represent the optimal Q-function — which is an $|\mathcal{S}||\mathcal{A}|H$-dimensional object — via $H$ parameter vectors each of dimension $d$. Throughout this work, an MDP obeying this condition is said to be *an MDP with linearly realizable optimal Q-function*, or more concisely, *an MDP with linear $Q^\star$*.

## 1.2 Sample size barriers with linearly realizable $Q^\star$

This paper focuses on MDPs with linearly realizable optimal Q-function $Q^\star$. In stark contrast to linear MDPs that allow for sample-efficient RL (in the sense that the sample complexity is almost independent of $|\mathcal{S}|$ and $|\mathcal{A}|$ but instead depends only polynomially on $d, H$ and possibly a certain sub-optimality gap), MDPs with linear $Q^\star$ do not admit a similar level of sample efficiency in general. To facilitate discussion, we summarize key existing results for a couple of settings. Here and below, $f(d) = \Omega(g(d))$ means that $f(d)$ is at least on the same order as $g(d)$ for $d$ large enough.

- *Sample inefficiency under a generative model.* Even when a generative model or a simulator is available — so that the learner can query arbitrary state-action pairs to draw samples from (Kearns and Singh, 1999) — one can find a hard MDP instance in this class that requires at least $\min\big\{\exp(\Omega(d)), \exp(\Omega(H))\big\}$ samples regardless of the algorithm in use (Weisz et al., 2021b).

- *Sample efficiency with a sub-optimality gap under a generative model.* The aforementioned sample size barrier can be alleviated if, for each state, there exists a sub-optimality gap $\Delta_{\mathsf{gap}}$ between the value under the optimal action and that under any sub-optimal action. As asserted by Du et al. (2020a, Appendix C), a sample size that scales polynomially in $d, H$ and $1/\Delta_{\mathsf{gap}}$ is sufficient to identify the optimal policy, assuming access to the generative model.

- *Sample inefficiency with a large sub-optimality gap in online RL.* Turning to the standard episodic online RL setting (so that in each episode, the learner is given an initial state and executes the MDP for $H$ steps to obtain a sample trajectory), sample-efficient algorithms are, much to our surprise, infeasible even in the presence of a large sub-optimality gap. As has been formalized in Wang et al. (2021b), it is possible to construct a hard MDP instance with a constant sub-optimality gap that cannot be solved without at least $\min\big\{\exp(\Omega(d)), \exp(\Omega(H))\big\}$ samples.

In conclusion, the linear $Q^\star$ assumption, while succinctly capturing the low-dimensional structure, still presents an undesirable hurdle for RL. The sampling mechanism commonly studied in RL literature precludes sample-efficient solutions even when a favorable sub-optimality gap exists.

## 1.3 Our contributions

Having observed the exponential separation between the generative model and standard online RL when it comes to the linear $Q^\star$ problem, one might naturally wonder whether there exist practically relevant sampling mechanisms — more flexible than standard online RL yet more practical than the generative model — that promise substantial sample size reduction. This motivates the investigation of the current paper, as summarized below.

*A new sampling protocol: sampling with state revisiting.* We investigate a flexible sampling protocol that is built upon classical online/exploratory formalism but allows for *state revisiting* (detailed

in Algorithm 1). In each episode, the learner starts by running the MDP for $H$ steps, and is then allowed to revisit any previously visited state and re-run the MDP from there. The learner is allowed to revisit states for an arbitrary number of times, although executing this feature too often might inevitably incur an overly large sampling burden. This sampling protocol accommodates several realistic scenarios; for instance, it captures the "save files" feature in video games that allows players to record player progress and resume from the save points later on. In addition, state revisiting is reminiscent of Monte Carlo Tree Search implemented in various real-world applications, which assumes that the learner can go back to father nodes (i.e., previous states) (Silver et al., 2016). This protocol is also referred to as *local access to the simulator* in the recent work Yin et al. (2021).

*A sample-efficient algorithm.* Focusing on the above sampling protocol, we propose a value-based method — called LinQ-LSVI-UCB — adapted from the LSVI-UCB algorithm (Jin et al., 2020). The algorithm implements the optimism principle in the face of uncertainty, while harnessing the knowledge of the sub-optimality gap to determine whether to revisit states. Our algorithm achieves a sample complexity that scales polynomially in the feature dimension $d$, the horizon $H$, and the inverse sub-optimality gap $1/\Delta_{\mathsf{gap}}$, but is otherwise independent of the size of the state/action space.

## 2 Model and assumptions

We present precise problem formulation as well as a couple of key assumptions. Here and throughout, we denote by $|\mathcal{S}|$ the cardinality of a set $\mathcal{S}$, and adopt the notation $[n] := \{1, \cdots, n\}$.

### 2.1 Basics of Markov decision processes

**Finite-horizon MDP.** The focus of this paper is the setting of a finite-horizon MDP, as represented by the quintuple $\mathcal{M} = (\mathcal{S}, \mathcal{A}, \{P_h\}_{h=1}^H, \{r_h\}_{h=1}^H, H)$. Here, $\mathcal{S} := \{1, \cdots, |\mathcal{S}|\}$ denotes the state space, $\mathcal{A} := \{1, \cdots, |\mathcal{A}|\}$ is the action space, $H$ is the time horizon, $P_h$ stands for the probability transition kernel at time step $h \in [H]$ (namely, $P_h(\cdot \mid s, a)$ is the transition probability from state $s$ upon execution of action $a$ at step $h$), whereas $r_h : \mathcal{S} \times \mathcal{A} \to [0, 1]$ represents the reward function at step $h$ (namely, we denote by $r_h(s, a)$ the immediate reward received at step $h$ when the current state is $s$ and the current action is $a$). For simplicity, it is assumed throughout that all rewards $\{r_h(s, a)\}$ are deterministic and reside within the range $[0, 1]$. Note that our analysis can also be directly extended to accommodate random rewards, which we omit here for the sake of brevity.

**Policy, value function, and Q-function.** We let $\pi = \{\pi_h\}_{1 \le h \le H}$ represent a policy or action selection rule. For each time step $h$, $\pi_h$ represents a deterministic mapping from $\mathcal{S}$ to $\mathcal{A}$, namely, action $\pi_h(s)$ is taken at step $h$ if the current state is $s$. The value function associated with policy $\pi$ at step $h$ is then defined as the cumulative reward received between steps $h$ and $H$ under this policy:

$$\forall (s, h) \in \mathcal{S} \times [H]: \qquad V_h^\pi(s) := \mathbb{E}\left[\sum_{t=h}^H r_t(s_t, a_t) \,\Big|\, s_t = s\right]. \tag{3}$$

Here, the expectation is taken over the randomness of an MDP trajectory $\{s_t\}_{t=h}^H$ induced by policy $\pi$ (namely, $a_t = \pi_t(s_t)$ and $s_{t+1} \sim P(\cdot \mid s_t, a_t)$ for any $h \le t \le H$). Similarly, the action-value function (or Q-function) associated with policy $\pi$ is defined as

$$\forall (s, a, h) \in \mathcal{S} \times \mathcal{A} \times [H]: \qquad Q_h^\pi(s, a) := \mathbb{E}\left[\sum_{t=h}^H r_t(s_t, a_t) \,\Big|\, s_h = s, a_h = a\right], \tag{4}$$

which resembles the definition (3) except that the action at step $h$ is frozen to be $a$. Our normalized reward assumption (i.e., $r_h(s, a) \in [0, 1]$) immediately leads to the trivial bounds

$$\forall (s, a, h) \in \mathcal{S} \times \mathcal{A} \times [H]: \qquad 0 \le V_h^\pi(s) \le H \qquad \text{and} \qquad 0 \le Q_h^\pi(s, a) \le H. \tag{5}$$

A recurring goal in reinforcement learning is to search for a policy that maximizes the value function and the Q-function. For notational simplicity, we define the optimal value function $V^\star = \{V_h^\star\}_{1 \le h \le H}$ and optimal Q-function $Q^\star = \{Q_h^\star\}_{1 \le h \le H}$ respectively as follows

$$\forall (s, a, h) \in \mathcal{S} \times \mathcal{A} \times [H]: \qquad V_h^\star(s) := \max_\pi V_h^\pi(s) \qquad \text{and} \qquad Q_h^\star(s, a) := \max_\pi Q_h^\pi(s, a),$$

with the optimal policy (i.e., the one that maximizes $V^\pi$) represented by $\pi^\star = \{\pi_h^\star\}_{1 \le h \le H}$.

## 2.2  Key assumptions

**Linear realizability of $Q^\star$.**  In order to enable significant reduction of sample complexity, it is crucial to exploit proper low-dimensional structure of the problem. This paper is built upon linear realizability of the optimal Q-function $Q^\star$ as follows.

**Assumption 1.**  *Suppose that there exist a collection of pre-determined feature maps*

$$\varphi = (\varphi_h)_{1 \le h \le H}, \qquad \varphi_h : \mathcal{S} \times \mathcal{A} \to \mathbb{R}^d \tag{6}$$

*and a set of unknown vectors $\theta_h^\star \in \mathbb{R}^d$ ($1 \le h \le H$) such that*

$$\forall (s, a, h) \in \mathcal{S} \times \mathcal{A} \times [H] : \qquad Q_h^\star(s, a) = \langle \varphi_h(s, a), \theta_h^\star \rangle. \tag{7}$$

*In addition, we assume that*

$$\forall (s, a, h) \in \mathcal{S} \times \mathcal{A} \times [H] : \qquad \|\varphi_h(s, a)\|_2 \le 1 \quad \text{and} \quad \|\theta_h^\star\|_2 \le 2H\sqrt{d}. \tag{8}$$

In other words, we assume that $Q^\star = \{Q_h^\star\}_{1 \le h \le H}$ can be embedded into a $d$-dimensional subspace encoded by $\varphi$, with $d \le |\mathcal{S}||\mathcal{A}|$. In fact, we shall often view $d$ as being substantially smaller than the ambient dimension $|\mathcal{S}||\mathcal{A}|$ in order to capture the dramatic degree of potential dimension reduction. It is noteworthy that linear realizability of $Q^\star$ in itself is a considerably weaker assumption compared to the one commonly assumed for linear MDPs (Jin et al., 2020) (which assumes $\{P_h\}_{1 \le h \le H}$ and $\{r_h\}_{1 \le h \le H}$ are all linearly parameterized). The latter necessarily implies the former, while in contrast the former by no means implies the latter. Additionally, we remark that the assumption (8) is compatible with what is commonly assumed for linear MDPs; see, e.g., Jin et al. (2020, Lemma B.1) for a reasoning about why this bound makes sense.

**Sub-optimality gap.**  As alluded to previously, another metric that comes into play in our theoretical development is the sub-optimality gap. Specifically, for each state $s$ and each time step $h$, we define the following metric

$$\Delta_h(s) := \min_{a \notin \mathcal{A}_s^\star} \left\{ V_h^\star(s) - Q_h^\star(s, a) \right\} \qquad \text{with } \mathcal{A}_s^\star := \left\{ a : Q_h^\star(s, a) = V_h^\star(s) \right\}. \tag{9}$$

In words, $\Delta_h(s)$ quantifies the gap — in terms of the resulting Q-values — between the optimal action and the sub-optimal ones. It is worth noting that there might exist multiple optimal actions for a given $(s, h)$ pair, namely, the set $\mathcal{A}_s^\star$ is not necessarily a singleton. Further, we define the minimum gap over all $(s, h)$ pairs as follows

$$\Delta_{\text{gap}} := \min_{s, h \in \mathcal{S} \times [H]} \Delta_h(s), \tag{10}$$

and refer to it as the sub-optimality gap throughout this paper.

## 2.3  RL under sampling with state revisiting

In standard online episodic RL settings, the learner collects data samples by executing multiple length-$H$ trajectories in the MDP $\mathcal{M}$ via suitably chosen policies; more concretely, in the $n$-th episode with a given initial state $s_0^n$, the agent executes a policy to generate a sample trajectory $\{(s_h^n, a_h^n)\}_{1 \le h \le H}$, where $(s_h^n, a_h^n)$ denotes the state-action pair at time step $h$. This setting underscores the importance of trading off exploitation and exploration. As pointed out previously, however, this classical sampling mechanism could be highly inefficient for MDPs with linearly realizable $Q^\star$, even in the face of a constant sub-optimality gap (Wang et al., 2021b).

**A new sampling protocol with state revisiting.**  In order to circumvent this sample complexity barrier, the current paper studies a more flexible sampling mechanism that allows one to revisit previous states in the same episode. Concretely, in each episode, the sampling process can be carried out in the following fashion:

---

**Algorithm 1:** Sampling protocol for an episode with state revisiting.

---
**1** **Input:** initial state $s_1$.

**2** Select a policy and sample a length-$H$ trajectory $\{(s_t, a_t)\}_{1 \le t \le H}$.

**3** **repeat**

**4**     Pick any previously visited state $s_h$ in this episode;

**5**     Execute a new trajectory starting from $s_h$ all the way up to step $H$, namely, $\{(s_t, a_t)\}_{h \le t \le H}$;
    here, we overload notation to simplify presentation.

**6** **until** *the learner terminates it.*

---

As a distinguishing feature, the sampling mechanism described in Algorithm 1 allows one to revisit previous states and retake samples from there, which reveals more information regarding these states. To make apparent its practice relevance, we first note that the generative model proposed in Kearns and Singh (1999); Kakade (2003) — in which one can query a simulator with arbitrary state-action pairs to get samples — is trivially subsumed as a special case of this sampling mechanism. Moving on to a more complicated yet realistic scenario, consider role-playing video games which commonly include built-in "save files" features. This type of features allows the player to record its progress at any given point, so that it can resume the game from this save point later on. In fact, rebooting the game multiple times from a saved point allows an RL algorithm to conduct trial-and-error learning for this particular game point.

As a worthy note, while revisiting a particular state many times certainly yields information gain about this state, it also means that fewer samples can be allocated to other episodes if the total sampling budget is fixed. Consequently, how to design intelligent state revisiting schemes in order to optimize sample efficiency requires careful thinking.

**Learning protocol and sample efficiency.** We are now ready to describe the learning process — which consists of $N$ episodes — and our goal.

- In the $n$-th episode ($1 \le n \le N$), the learner is given an initial state $s_1^{(n)}$ (assigned by nature), and executes the sampling protocol in Algorithm 1 until this episode is terminated.

- At the end of the $n$-th episode, the outcome of the learning process takes the form of a policy $\pi^{(n)}$, which is learned based on all information collected up to the end of this episode.

The quality of the learning outcome $\{\pi^{(n)}\}_{1 \le n \le N}$ is then measured by the cumulative regret over $N$ episodes as follows:

$$\mathsf{Regret}(N) := \sum_{n=1}^{N} \left( V_1^\star\big(s_1^{(n)}\big) - V_1^{\pi^{(n)}}\big(s_1^{(n)}\big) \right), \tag{11}$$

which is what we aim to minimize under a given sampling budget. More specifically, for any target level $\varepsilon \in [0, H]$, the aim is to achieve

$$\frac{1}{N}\mathsf{Regret}(N) \le \varepsilon$$

regardless of the initial states (which are chosen by nature), using a sample size $T$ no larger than $\mathsf{poly}\big(d, H, \frac{1}{\varepsilon}, \frac{1}{\Delta_{\mathsf{gap}}}\big)$ (but independent of $|\mathcal{S}|$ and $|\mathcal{A}|$). Here and throughout, $T$ stands for the total number of samples observed in the learning process; for instance, a new trajectory $\{(s_t, a_t)\}_{h \le t \le H}$ amounts to $H - h$ new samples. Due to the presence of state revisiting, there is a difference between our notions of regret / sample complexity and the ones used in standard online RL, which we shall elaborate on in the next section. An RL algorithm capable of achieving this level of sample complexity is declared to be sample-efficient, given that the sample complexity does not scale with the ambient dimension of the problem (which could be enormous in contemporary RL).

***Remark*** 1 (From average regret to PAC guarantees and optimal policies.). There is some intimate connection between regret bounds and PAC guarantees that has been pointed out previously (e.g., Jin et al. (2018)). For instance, by fixing the initial state distribution to be identical (e.g., $s_1^{(n)} = s$ for all $1 \le n \le N$) and choosing the output policy $\widehat{\pi}$ uniformly at random from $\{\pi^{(n)} \mid 1 \le n \le N\}$, one can easily verify that this output policy $\widehat{\pi}$ is $\varepsilon$-optimal for state $s$, as long as $\frac{1}{N}\mathsf{Regret}(N) \le \varepsilon$.

# 3 Algorithm and main results

In this section, we put forward an algorithm tailored to the sampling protocol described in Algorithm 1, and demonstrate its desired sample efficiency.

## 3.1 Algorithm

Our algorithm design is motivated by the method proposed in (Jin et al., 2020) for linear MDPs — called *least-squares value iteration with upper confidence bounds (LSVI-UCB)* — which follows the principle of "optimism in the face of uncertainty". In what follows, we shall begin by briefly reviewing the key update rules of LSVI-UCB, and then discuss how to adapt it to accommodate MDPs with linearly realizable $Q^\star$ when state revisiting is permitted.

**Review: LSVI-UCB for linear MDPs.** Let us remind the readers of the linear MDP setting. We assume that there exist unknown vectors $\mu_h(\cdot) = [\mu_h^{(1)}, \cdots, \mu_h^{(d)}]^\top \in \mathbb{R}^{d \times |\mathcal{S}|}$ and $w_h \in \mathbb{R}^d$ such that

$$\forall (s, a, h) \in \mathcal{S} \times \mathcal{A} \times [H]: \quad P_h(\cdot \mid s, a) = \big\langle \varphi(s, a), \mu_h(\cdot) \big\rangle \ \text{ and } \ r_h(s, a) = \big\langle \varphi(s, a), w_h(s, a) \big\rangle.$$

In other words, both the probability transition kernel and the reward function can be linearly represented using the set of feature maps $\{\varphi(s, a)\}$.

LSVI-UCB can be viewed as a generalization of the UCBVI algorithm (Azar et al., 2017) (originally proposed for the tabular setting) to accommodate linear function approximation. In each episode, the learner draws a sample trajectory following the greedy policy w.r.t. the current Q-function estimate with UCB exploration; namely, an MDP trajectory $\{(s_h^n, a_h^n)\}_{1 \le h \le H}$ is observed in the $n$-th episode. Working backwards (namely, going from step $H$ all the way back to step 1), the LSVI-UCB algorithm in the $n$-th episode consists of the following key updates:

$$\Lambda_h \ \leftarrow \ \sum_{i=1}^n \varphi(s_h^i, a_h^i) \varphi(s_h^i, a_h^i)^\top + \lambda I, \tag{12a}$$

$$\theta_h \ \leftarrow \ \Lambda_h^{-1} \sum_{i=1}^n \varphi(s_h^i, a_h^i) \Big\{ r_h(s_h^i, a_h^i) + \max_a Q_{h+1}(s_{h+1}^i, a) \Big\}, \tag{12b}$$

$$Q_h(\cdot, \cdot) \ \leftarrow \ \min \left\{ \big\langle \theta_h, \varphi(\cdot, \cdot) \big\rangle + \beta \sqrt{\varphi(\cdot, \cdot)^\top \Lambda_h^{-1} \varphi(\cdot, \cdot)}, H \right\}, \tag{12c}$$

with the regularization parameter $\lambda$ set to be 1. Informally speaking, $\theta_h$ (cf. (12b)) corresponds to the solution to a ridge-regularized least-squares problem — tailored to solving the Bellman optimality equation with linear parameterization — using all samples collected so far for step $h$, whereas the matrix $\Lambda_h$ (cf. (12a)) captures the (properly regularized) covariance of $\varphi(\cdot, \cdot)$ associated with these samples. In particular, $\big\langle \theta_h, \varphi(\cdot, \cdot) \big\rangle$ attempts to estimate the Q-function by exploiting its linear representation, and the algorithm augments it by an upper confidence bound (UCB) bonus $\beta \sqrt{\varphi(\cdot, \cdot)^\top \Lambda_h^{-1} \varphi(\cdot, \cdot)}$ — a term motivated by the linear bandit literature (Lattimore and Szepesvári, 2020) — to promote exploration, where $\beta$ is a hyper-parameter to control the level of exploration. The update rule (12c) also ensures that the Q-estimate never exceeds the trivial upper bound $H$.

**Our algorithm: LinQ-LSVI-UCB for linearly realizable $Q^\star$.** Moving from linear MDPs to MDPs with linear $Q^\star$, we need to make proper modification of the algorithm. To facilitate discussion, let us introduce some helpful concepts.

- Whenever we start a new episode or revisit a state (and draw samples thereafter), we say that *a new path* is being collected. The total number of paths we have collected is denoted by $K$.

- For each $k$ and each step $h$, we define a set of indices

$$\mathcal{I}_h^k := \big\{ i : \ 1 \le i \le k \mid \theta_h^i \text{ is updated in the } i\text{-th path at time step } h \big\}, \tag{13}$$

which will be described precisely in Algorithm 2. As we shall see, the cardinality of $\mathcal{I}_h^k$ is equal to the total number of new samples that have been collected at time step $h$ up to the $k$-th path.

We are now ready to describe the algorithm. For the $k$-th path, our algorithm proceeds as follows.

- *Sampling.* Suppose that we start from a state $s_h^k$ at time step $h$. The learner adopts the greedy policy $\pi^k = \{\pi_j^k\}_{h \leq j \leq H}$ in accordance with the current Q-estimate $\{Q_j^{k-1}\}_{h \leq j \leq H}$, and observes a fresh sample trajectory $\{(s_j^k, a_j^k)\}_{h \leq j \leq H}$ as follows: for $j = h, h+1, \ldots, H$,

$$s_{j+1}^k \sim P_j(\cdot \,|\, s_j^k, a_j^k) \qquad \text{with} \qquad a_j^k = \pi_j^k(s_j^k) := \arg\max_a Q_j^{k-1}(s_j^k, a). \tag{14}$$

- *Backtrack and update estimates.* We then work backwards to update our Q-estimates and the $\theta$-estimates (i.e., estimates for the linear representation of $Q^\star$), until the UCB bonus term (which reflects the estimated uncertainty level of the Q-estimate) drops below a threshold determined by the sub-optimality gap $\Delta_{\mathsf{gap}}$. More precisely, working backwards from $h = H$, we carry out the following calculations if certain conditions (to be described shortly) are met:

$$\Lambda_h^k \;\leftarrow\; \sum_{i \in \mathcal{I}_h^k} \varphi_h(s_h^i, a_h^i)\varphi_h(s_h^i, a_h^i)^\top + I, \tag{15a}$$

$$\theta_h^k \;\leftarrow\; \left(\Lambda_h^k\right)^{-1} \sum_{i \in \mathcal{I}_h^k} \varphi_h(s_h^i, a_h^i)\Big\{r_h(s_h^i, a_h^i) + \big\langle \varphi_{h+1}(s_{h+1}^i, a_{h+1}^i), \theta_{h+1}^k \big\rangle\Big\}, \tag{15b}$$

$$b_h^k(\cdot, \cdot) \;\leftarrow\; \beta\sqrt{\varphi_h(\cdot, \cdot)^\top \left(\Lambda_h^k\right)^{-1}\varphi_h(\cdot, \cdot)}, \tag{15c}$$

$$Q_h^k(\cdot, \cdot) \;\leftarrow\; \min\Big\{\big\langle \varphi_h(\cdot, \cdot), \theta_h^k \big\rangle + b_h^k(\cdot, \cdot), \, H\Big\}. \tag{15d}$$

Here, we employ the pre-factor

$$\beta = c_\beta \sqrt{dH^4 \log \frac{KH}{\delta}} \tag{16}$$

to adjust the level of "optimism", where $c_\beta > 0$ is some suitably large constant. Crucially, whether the update (15b) — and hence (15d) — is executed depends on the size of the bonus term $b_{h+1}^{k-1}$ of the last attempt at step $h+1$. Informally, if the bonus term is sufficiently small compared to the sub-optimality gap, then we have confidence that the policy estimate (after time step $h$) can be trusted in the sense that it is guaranteed to generalize and perform well on unseen states.

The complete algorithm is summarized in Algorithm 2, with some basic auxiliary functions provided in Algorithm 3. To facilitate understanding, an illustration is provided in Figure 1.

Two immediate remarks are in order. In comparison to LSVI-UCB in Jin et al. (2020), the update rule (15b) for $\theta_h^k$ employs the linear representation $\big\langle \varphi_{h+1}(s_{h+1}^i, a_{h+1}^i), \theta_{h+1}^k \big\rangle$ *without the UCB bonus* as the Q-value estimate. This subtle difference turns out to be important in the analysis for MDPs with linear $Q^\star$. In addition, LSVI-UCB is equivalent to first obtaining a linear representation of transition kernel $P$ (Agarwal et al., 2019) and then using it to build Q-function estimates and draw samples. In contrast, Algorithm 2 cannot be interpreted as a decoupling of model estimation and planning/exploration stage, and is intrinsically a value-based approach.

## 3.2 Theoretical guarantees

Equipped with the precise description of Algorithm 2, we are now positioned to present its regret bound and sample complexity analysis. Our result is this:

**Theorem 1.** *Suppose that Assumption 1 holds, and that $c_\beta \geq 8$ is some fixed constant. Then for any $0 < \delta < 1$, and any initial states $\big\{s_1^{(n)}\big\}_{1 \leq n \leq N}$, Algorithm 2 achieves a regret (see (11)) obeying*

$$\frac{1}{N}\mathsf{Regret}(N) \leq 8c_\beta \sqrt{\frac{d^2 H^7 \log^2 \frac{HT}{\delta}}{T}} \tag{17}$$

*with probability at least $1 - \delta$, provided that $N \geq \frac{4c_\beta^2 d^2 H^5 \log^2 \frac{HT}{\delta}}{\Delta_{\mathsf{gap}}^2}$. In addition, the total number of state revisits satisfies*

$$K - N \leq \frac{4c_\beta^2 d^2 H^5 \log^2 \frac{KH}{\delta}}{\Delta_{\mathsf{gap}}^2}. \tag{18}$$

---

**Algorithm 2:** LinQ-LSVI-UCB with state revisiting.

1 **inputs:** number of episodes $N$, sub-optimality gap $\Delta_{\mathsf{gap}}$, initial states $\{s_1^{(n)}\}_{1 \leq n \leq N}$.
2 **initialization:** $h = 0$, $n = 0$, $\theta_j^0 = 0$ and $\mathcal{I}_j^0 = \emptyset$ for all $1 \leq j \leq H$.
3 **for** $k = 1, 2, \cdots$ **do**
4   call $\pi^k \leftarrow$ get-policy().
5   $h \leftarrow h + 1$.
6   **if** $h = 1$ **then**
7    $\pi^{(n)} \leftarrow \pi^{k-1}$. // record the up-to-date policy learned in this episode.
8    $n \leftarrow n + 1$. // start a new episode.
9    **if** $n > N$ **then**
10     $K \leftarrow k - 1$ and **return**. // terminate after $N$ episodes.
11    Set the initial state $s_1^k = s_1^{(n)}$.
12   call sampling(). // collect new samples from step $h$; see Algorithm 3.
   /* backtrack and determine whether to revisit a state and redraw new samples.           */
13   Set $\theta_{H+1}^k = 0$ and $h = H$.
14   **while** $h > 0$ *and* $b_{h+1}^{k-1}(s_{h+1}^k, a_{h+1}^k) < \Delta_{\mathsf{gap}}/2$ *(cf. (15c))* **do**
15    $\mathcal{I}_h^k \leftarrow \mathcal{I}_h^{k-1} \bigcup \{k\}$. // expand $\mathcal{I}_h^k$ whenever we need to update $\theta_h^k$.
16    Update $\theta_h^k$ according to (15b).
17    $h \leftarrow h - 1$.
18   call update-remaining(). // keep remaining iterates unchanged; see Algorithm 3.

---

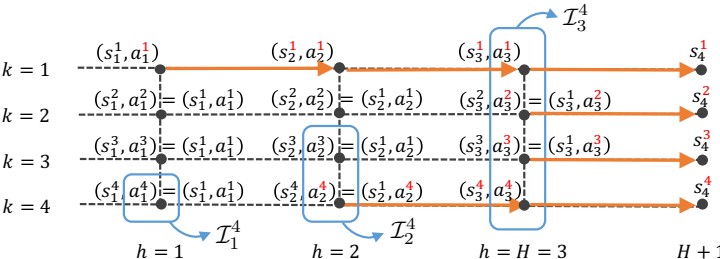

Figure 1: Illustration of $\mathcal{I}_h^K$ for a simple scenario where $N = 1$, $H = 3$, and the number of paths is $K = 4$. After the 1st path, the sampling process revisits state $s_3^1$ twice (each time drawing one new sample), and then revisits state $s_2^1$ to draw two samples from there. This episode then terminates as the conditions are met for all steps. A red superscript indicates new state or new actions, and the orange line illustrates the sampling process. Here, $\mathcal{I}_3^4 = \{1, 2, 3, 4\}$, $\mathcal{I}_2^4 = \{3, 4\}$ and $\mathcal{I}_1^4 = \{4\}$, which record the paths where the linear representations are updated for each respective step.

Theorem 1 characterizes the sample efficiency of the proposed algorithm. Specifically, the theorem implies that for any $0 < \varepsilon < H$, the average regret satisfies $\frac{1}{N}\mathsf{Regret}(N) \leq \varepsilon$ with probability exceeding $1 - \delta$, once the total number $T$ of samples exceeds

$$T \geq \frac{64 c_\beta^2 d^2 H^7 \log^2 \frac{HT}{\delta}}{\varepsilon^2}. \tag{19}$$

Several implications and further discussions of this result are in order.

**Efficiency of the proposed algorithm.** We highlight several benefits of our algorithm.

- *Sample efficiency.* While our sample complexity bound (19) scales as a polynomial function of both $d$ and $H$, it does not rely on either $|\mathcal{S}|$ or $|\mathcal{A}|$. This hints at the dramatic sample size reduction when $|\mathcal{S}||\mathcal{A}|$ far exceeds the feature dimension $d$ and the horizon $H$.

---

**Algorithm 3:** Simple auxiliary functions.

---

**1** **Function** `sampling()`:

    /* sampling from the beginning (if $h = 0$) or from a revisited state (if $h > 0$).     */

    /* do not update samples prior to step $h$.     */

**2**     **for** $j = 1, 2, \ldots, h - 1$ **do**

**3**         Set $a_j^k = a_j^{k-1}$, and $s_{j+1}^k = s_{j+1}^{k-1}$.

    /* a new round of sampling from step $h$.     */

**4**     **for** $j = h, h + 1, \cdots, H$ **do**

**5**         Compute $Q_j^{k-1}(s_j^k, a)$ according to (15d).

**6**         Take $a_j^k = \arg\max_a Q_j^{k-1}(s_j^k, a)$, and draw $s_{j+1}^k \sim P_j(\cdot \mid s_j^k, a_j^k)$.

**7** **Function** `update-remaining()`:

    /* keep the estimates prior to step $h$ unchanged.     */

**8**     **for** $j = 1, 2, \ldots, h$ **do**

**9**         Set $\theta_j^k = \theta_j^{k-1}$.

**10** **Function** `get-policy()`:

    /* update the Q-estimates.     */

**11**     **for** $1 \le h \le H$ **do**

**12**         Set $Q_h^{k-1}(\cdot, \cdot)$ according to (15d).

**13**         $\pi_h^k(\cdot) \leftarrow \arg\max_a Q_h^{k-1}(\cdot, a)$.

---

- *A small number of state revisits.* Theorem 1 develops an upper bound (18) on the total number of state revisits, which is gap-dependent but otherwise independent of the target accuracy level $\varepsilon$. As a consequence, as the sample size $T$ increases (or when $\varepsilon$ decreases), the ratio of the number of state revisits to the sample size becomes vanishingly small, meaning that the true sampling process in our algorithm becomes increasingly closer to the standard online RL setting.

- *Computational complexity and memory complexity.* The computational bottleneck of the proposed algorithm lies in the update of $\theta_h^k$ and $b_h^k$ (see (15b) and (15c), respectively), which consists of solving a linear systems of equations and can be accomplished using, say, the conjugate gradient method in time on the order of $d^2$ (up to logarithmic factor). In addition, one needs to search over all actions when drawing samples, so the algorithm necessarily depends on $|\mathcal{A}|$. In total, the algorithm has a runtime no larger than $\widetilde{O}(d^2|\mathcal{A}|T)$, and requires $O(d^2 H)$ units of memory.

**Cumulative regret over paths.** Due to the introduction of state revisiting, there are two possible ways to accumulate regrets: over the episodes or over the paths. While our analysis so far adopts the former (see (11)), it is not difficult to translate our regret bound over the episodes to the one over the paths. To be more precise, let us denote the regret over paths as follows for distinguishing purposes:

$$\mathsf{Regret}_{\mathsf{path}}(K) := \sum_{k=1}^{K} \left( V_1^\star(s_1^k) - V_1^{\pi^k}(s_1^k) \right). \tag{20}$$

A close inspection of our analysis readily reveals the following regret upper bound

$$\mathsf{Regret}_{\mathsf{path}}(K) \le 4c_\beta \sqrt{d^2 H^6 K \log^2 \frac{HT}{\delta}} + \frac{4c_\beta^2 d^2 H^6 \log^2 \frac{KH}{\delta}}{\Delta_{\mathsf{gap}}^2} \tag{21a}$$

with probability exceeding $1 - \delta$; see Section B.2 for details. This bound confirms that the regret over the paths exhibits a scaling of at most $\sqrt{K}$.

**Logarithmic regret.** Our analysis further leads to a significantly strengthened upper bound on the regret. As we shall solidify in Section B.2, the regret incurred by our algorithm satisfies

$$\mathbb{E}\big[\mathsf{Regret}(N)\big] \le \mathbb{E}\big[\mathsf{Regret}_{\mathsf{path}}(K)\big] \le \frac{17c_\beta^2 d^2 H^7 \log^2(KH)}{\Delta_{\mathsf{gap}}^2}, \tag{21b}$$

largely owing to the presence of the gap assumption. This implies that the expected regret scales only logarithmically in the number of paths $K$, which could often be much smaller than the previous bound (21a). In fact, this is consistent with the recent literature regarding logarithmic regrets under suitable gap assumptions (e.g., Simchowitz and Jamieson (2019); Yang et al. (2021) for tabular MDPs and He et al. (2021) for the case with linear function approximation).

**Comparison to the case with a generative model.**   We find it helpful to compare our findings with the algorithm developed in the presence of a generative model. In a nutshell, the algorithm described in Du et al. (2020a, Appendix C) starts by identifying a "well-behaved" basis of the feature vectors, and then queries the generative model to sample the state-action pairs related to this basis. In contrast, our sampling protocol (cf. Algorithm 1) is substantially more restrictive and does not give us the freedom to sample such a basis. In fact, our algorithm is exploratory in nature, which is more challenging to analyze than the case with a generative model.

We shall also point out a technical difference between our approach and the algorithm in Du et al. (2020a). A key insight in Du et al. (2020a) is that: by sampling each *anchor* state-action pair for $\mathsf{poly}(1/\Delta_{\mathsf{gap}})$ times, one can guarantee sufficiently accurate Q-estimates in all state-action pairs, which in turn ensures $\pi_k = \pi^\star$ in all future estimates. This, however, is not guaranteed in our algorithm when it comes to the state revisiting setting. Fortunately, the gap condition helps ensure that there are at most $\mathsf{poly}(1/\Delta_{\mathsf{gap}})$ number of samples such that $\pi_k \neq \pi^\star$, although the discrepancy might happen at any time throughout the execution of the algorithm (rather than only happening at the beginning). In addition, careful use of state revisiting helps avoid these sub-optimal estimates by resetting for at most $\mathsf{poly}(1/\Delta_{\mathsf{gap}})$ times, which effectively prevents error blowup.

**Comparison to prior works under state revisiting.**   Upon closer examination, the sampling mechanism of Weisz et al. (2021a) considers another kind of state revisiting strategy and turns out to be quite similar to ours, which accesses a batch of samples $\{(s_h, a_h, s_{h+1}^i)\}_{i \geq 1}$ for the current state $s_h$ with all actions $a_h \in \mathcal{A}$. where $s_{h+1}^i$ represents the $i$-th attempt to draw sample transition from $s_h$. Assuming only $V^\star$ is linearly realizable, their sample complexity is on the order of $(dH)^{|\mathcal{A}|}$, and hence its sample efficiency depends highly on the condition that $\mathcal{A} = O(1)$. Additionally, Du et al. (2020b) proposed an algorithm — tailored to a setting with deterministic transitions — that requires sampling each visited state multiple times (and hence can be accomplished when state revisiting is permitted); this algorithm might be extendable to accommodate stochastic transitions. Additional discussions of related works can be found in Section A in the supplementary material.

# 4   Discussion

In this paper, we have made progress towards understanding the plausibility of achieving sample-efficient RL when the optimal Q-function is linearly realizable. While prior works suggested an exponential sample size barrier in the standard online RL setting even in the presence of a constant sub-optimality gap, we demonstrate that this barrier can be conquered by permitting state revisiting (also called local access to generative models). An algorithm called LinQ-LSVI-UCB has been developed that provably enjoys a reduced sample complexity, which is polynomial in the feature dimension, the horizon and the inverse sub-optimality gap, but otherwise independent of the dimension of the state/action space.

Note, however, that linear function approximation for online RL remains a rich territory for further investigation. In contrast to the tabular setting, the feasibility and limitations of online RL might vary drastically across different families of linear function approximation. There are numerous directions that call for further theoretical development in order to obtain a more complete picture. For instance, can we identify other flexible, yet practically relevant, online RL sampling mechanisms that also allow for sample size reduction? Can we derive the information-theoretic sampling limits for various linear function approximation classes, and characterize the fundamental interplay between low-dimensional representation and sampling constraints? Moving beyond linear realizability assumptions, a very recent work Yin et al. (2021) showed that a gap-independent sample size reduction is feasible by assuming that $Q^\pi$ is linearly realizable for any policy $\pi$. However, what is the sample complexity limit for this class of function approximation remains largely unclear, particularly when state revisiting is not permitted. All of these are interesting questions for future studies.

## Acknowledgments and Disclosure of Funding

The authors are grateful to Csaba Szepesvári and Ruosong Wang for helpful discussions about Weisz et al. (2021a) and Du et al. (2019, 2020b), respectively. Y. Chen is supported in part by the grants AFOSR YIP award FA9550-19-1-0030, ONR N00014-19-1-2120, ARO YIP award W911NF-20-1-0097, NSF CCF-2106739, CCF-1907661, DMS-2014279 and IIS-1900140, and the Princeton SEAS Innovation Award. Y. Chi is supported in part by the grants ONR N00014-18-1-2142 and N00014-19-1-2404, and NSF CCF-2106778, CCF-1806154 and CCF-2007911. Y. Gu is supported in part by the grant NSFC-61971266. Y. Wei is supported in part by the grants NSF CCF-2106778, CCF-2007911 and DMS-2147546/2015447. Part of this work was done while Y. Chen and Y. Wei were visiting the Simons Institute for the Theory of Computing.

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
