# A  Additional related works

Non-asymptotic sample complexity guarantees for RL algorithms have been studied extensively in the tabular setting over recent years, e.g., Azar et al. (2013); Jaksch et al. (2010); Azar et al. (2017); Osband et al. (2016); Even-Dar and Mansour (2003); Dann and Brunskill (2015); Sidford et al. (2018); Zhang et al. (2020); Li et al. (2020a); Agarwal et al. (2020b); Yang et al. (2021); Li et al. (2020b, 2021b,a); Wainwright (2019); Agarwal et al. (2020c); Cen et al. (2020), which have been, to a large extent, well-understood. The sample complexity typically scales at least linearly with respect to the state space size $|\mathcal{S}|$ and the action space size $|\mathcal{A}|$, and therefore, falls short of being sample-efficient when the state/action space is of astronomical size. In contrast, theoretical investigation of RL with function approximations is still in its infancy due to the complicated interaction between the dynamic of the MDP with the function class. In fact, theoretical support remains highly inadequate even when it comes to linear function approximation. For example, plain Q-learning algorithms coupled with linear approximation might easily diverge (Baird, 1995). It is thus of paramount interest to investigate how to design algorithms that can efficiently exploit the low-dimensional structure without compromising learning accuracy. In what follows, we shall discuss some of the most relevant results to ours. The reader is also referred to the summaries of recent literature in Du et al. (2020a, 2021).

**Linear MDP.**  Yang and Wang (2019); Jin et al. (2020) proposed the linear MDP model, which can be regarded as a generalization of the linear bandit model (Abbasi-Yadkori et al., 2011; Dimakopoulou et al., 2019) and has attracted enormous recent activity (see e.g., Wang et al. (2019); Yang and Wang (2020); Zanette et al. (2020a); He et al. (2020); Du et al. (2020a); Wang et al. (2020a); Hao et al. (2020); Wang et al. (2021a); Wei et al. (2021); Touati and Vincent (2020) and the references therein). The results have further been generalized to scenarios with much larger feature dimension by exploiting proper kernel function approximation (Yang et al., 2020; Long and Han, 2021).

**From completeness to realizability.**  Du et al. (2020a) considered the policy completeness assumption, which assumes that the Q-functions of *all policies* reside within a function class that contains all functions that are linearly representable in a known low-dimensional feature space. In particular, Du et al. (2020a); Lattimore et al. (2020) examined how the model misspecification error propagates and impacts the sample efficiency of policy learning. A related line of works assumed that the linear function class is closed or has low approximation error under the Bellman operator, referred to as low inherent Bellman error (Munos, 2005; Shariff and Szepesvári, 2020; Zanette et al., 2019, 2020b).

These assumptions remain much stronger than the *realizability* assumption considered herein, where only the optimal Q-function $Q^\star$ is assumed to be linearly representable. Wen and Van Roy (2017); Du et al. (2020b) showed that sample-efficient RL is feasible in deterministic systems, which has been extended to stochastic systems with low variance in Du et al. (2019) under additional gap assumptions. In addition, Weisz et al. (2021b) established exponential sample complexity lower bounds under the generative model when only $Q^\star$ is linearly realizable; their construction critically relied on making the action set exponentially large. When restricted to a constant-size action space, Weisz et al. (2021a) provided a sample-efficient algorithm when only $V^\star$ is linearly realizable, where their sampling protocol essentially matches ours. Recently, Du et al. (2021) introduced the bilinear class and proposed sample-efficient algorithms when both $V^\star$ and $Q^\star$ are linearly realizable in the online setting.

**Beyond linear function approximation.**  Moving beyond linear function approximation, another line of works (Ayoub et al., 2020; Zhou et al., 2020) investigated mixtures of linear MDPs. Moreover, additional efforts have been dedicated to studying the low-rank MDP model (without knowing *a priori* the feature space), which aims to learn the low-dimensional features as part of the RL problem; partial examples include Agarwal et al. (2020a); Modi et al. (2021). We conclude by mentioning in passing other attempts in identifying tractable families of MDPs with structural assumptions, such as Jin et al. (2021); Jiang et al. (2017); Wang et al. (2020b); Osband and Van Roy (2014).

# B  Analysis

Before proceeding, we introduce several convenient notation to be used throughout the proof. As before, the total number of paths that have been sampled is denoted by $K$. For any $(s, a, h) \in$

$\mathcal{S} \times \mathcal{A} \times [H]$, we abbreviate

$$P_{h,s,a} := P_h(\cdot \mid s, a) \in \mathbb{R}^{|\mathcal{S}|}. \tag{22}$$

For any time step $h$ in the $k$-th path, we define the empirical distribution vector $P_h^k \in \mathbb{R}^{|\mathcal{S}|}$ such that

$$P_h^k(s) := \begin{cases} 1, & \text{if } s = s_{h+1}^k; \\ 0, & \text{if } s \neq s_{h+1}^k. \end{cases} \tag{23}$$

The value function estimate $V_h^k : \mathcal{S} \to \mathbb{R}$ at time step $h$ after observing the $k$-th path is defined as

$$\forall (s, h, k) \in \mathcal{S} \times [H] \times [K]: \qquad V_h^k(s) := \max_{a \in \mathcal{A}} Q_h^k(s, a), \tag{24}$$

where the iterate $Q_h^k$ is defined in (15d).

Further, we remind the reader the crucial notation $\mathcal{I}_h^k$ introduced in (13), which represents the set of paths between the 1st and the $k$-th paths that update the estimate of $\theta_h^\star$. We have the following basic facts.

**Lemma 1.** *For all $1 \leq k \leq K$ and $1 \leq h \leq H$, one has*

$$\mathcal{I}_h^k \subseteq \mathcal{I}_{h+1}^k. \tag{25}$$

*In addition,*

$$\left| \mathcal{I}_1^K \right| = N \qquad \text{and} \qquad \left| \mathcal{I}_H^K \right| = K. \tag{26}$$

*Proof.* This lemma is somewhat self-evident from our construction, and hence we only provide brief explanation. The first claim (25) holds true since if $\theta_h^i$ is updated in the $i$-th path, then $\theta_j^i$ $(j \geq h)$ must also be updated. The second claim (26) arises immediately from the definition of $N$ (i.e., the number of episodes) and $K$ (i.e., the number of paths). $\square$

### B.1 Main steps for proving Theorem 1

In order to bound the regret for our proposed estimate, we first make note of an elementary relation that follows immediately from our construction:

$$\sum_{n=1}^N \left( V_1^\star(s_1^{(n)}) - V_1^{\pi^{(n)}}(s_1^{(n)}) \right) = \sum_{k \in \mathcal{I}_1^K} \left( V_1^\star(s_1^k) - V_1^{\pi^k}(s_1^k) \right), \tag{27}$$

where we recall the definition of $\mathcal{I}_1^K$ in (13). It thus comes down to bounding the right-hand side of (27).

**Step 1: showing that $Q_h^k$ is an optimistic view of $Q_h^\star$.** Before proceeding, let us first develop a sandwich bound pertaining to the estimate error of the estimate $Q_h^k$ delivered by Algorithm 2. The proof of this result is postponed to Section C.1.

**Lemma 2.** *Suppose that $c_\beta \geq 8$. With probability at least $1 - \delta$, the following bound*

$$0 \leq Q_h^k(s, a) - Q_h^\star(s, a) \leq 2b_h^k(s, a) \tag{28}$$

*holds simultaneously for all $(s, a, k, h) \in \mathcal{S} \times \mathcal{A} \times [K] \times [H]$.*

In words, this lemma makes apparent that $Q_h^k$ is an over-estimate of $Q_h^\star$, with the estimation error dominated by the UCB bonus term $b_h^k$. This lemma forms the basis of the optimism principle.

**Step 2: bounding the term on the right-hand side of** (27). To control the difference $V_1^\star(s_1^k) - V_1^{\pi^k}(s_1^k)$ for each $k$, we establish the following two properties. First, combining Lemma 2 with the definition (24) (i.e., $V_h^{k-1}(s_h^k) = \max_a Q_h^{k-1}(s_h^k, a)$), one can easily see that

$$V_h^{k-1}(s_h^k) \geq Q_h^{k-1}(s_h^k, \pi^\star(s_h^k, h)) \geq Q_h^\star(s_h^k, \pi_h^\star(s_h^k)) = V_h^\star(s_h^k), \tag{29}$$

namely, $V_h^{k-1}$ is an over-estimate of $V_h^\star$. In addition, from the definition $a_h^k = \pi_h^k(s_h^k) = \arg\max_a Q_h^{k-1}(s_h^k, a)$, one can decompose the difference $V_h^{k-1}(s_h^k) - V_h^{\pi^k}(s_h^k)$ as follows

$$
\begin{aligned}
V_h^{k-1}(s_h^k) - V_h^{\pi^k}(s_h^k) &= Q_h^{k-1}(s_h^k, a_h^k) - Q_h^{\pi^k}(s_h^k, a_h^k) \\
&= Q_h^{k-1}(s_h^k, a_h^k) - Q_h^\star(s_h^k, a_h^k) + Q_h^\star(s_h^k, a_h^k) - Q_h^{\pi^k}(s_h^k, a_h^k) \\
&= Q_h^{k-1}(s_h^k, a_h^k) - Q_h^\star(s_h^k, a_h^k) + P_{h, s_h^k, a_h^k}(V_{h+1}^\star - V_{h+1}^{\pi^k}) \\
&= Q_h^{k-1}(s_h^k, a_h^k) - Q_h^\star(s_h^k, a_h^k) + (P_{h, s_h^k, a_h^k} - P_h^k)(V_{h+1}^\star - V_{h+1}^{\pi^k}) + V_{h+1}^\star(s_{h+1}^k) - V_{h+1}^{\pi^k}(s_{h+1}^k),
\end{aligned}
\tag{30}
$$

where the third line invokes Bellman equation $Q^\pi(s, a) = r(s, a) + P_{h, s, a} V^\pi$ for any $\pi$, and the last line makes use of the notation (23). Combining the above two properties leads to

$$
\begin{aligned}
\sum_{k \in \mathcal{I}_1^K} \left[ V_1^\star(s_1^k) - V_1^{\pi^k}(s_1^k) \right] &\le \sum_{k \in \mathcal{I}_1^K} \left[ V_1^{k-1}(s_1^k) - V_1^{\pi^k}(s_1^k) \right] \\
&= \sum_{k \in \mathcal{I}_1^K} \left[ V_2^\star(s_2^k) - V_2^{\pi^k}(s_2^k) \right] + \sum_{k \in \mathcal{I}_1^K} \left[ Q_1^{k-1}(s_1^k, a_1^k) - Q_1^\star(s_1^k, a_1^k) + (P_{1, s_1^k, a_1^k} - P_1^k)(V_2^\star - V_2^{\pi^k}) \right] \\
&\le \sum_{k \in \mathcal{I}_2^K} \left[ V_2^\star(s_2^k) - V_2^{\pi^k}(s_2^k) \right] + \sum_{k \in \mathcal{I}_1^K} \left[ Q_1^{k-1}(s_1^k, a_1^k) - Q_1^\star(s_1^k, a_1^k) + (P_{1, s_1^k, a_1^k} - P_1^k)(V_2^\star - V_2^{\pi^k}) \right],
\end{aligned}
$$

where the last line comes from the observation that $\mathcal{I}_h^K \subseteq \mathcal{I}_{h+1}^K$ (see Lemma 1). Applying the above relation recursively and using the fact that $V_{H+1}^\pi = 0$ for any $\pi$, we see that with probability at least $1 - \delta$,

$$
\begin{aligned}
\sum_{k \in \mathcal{I}_1^K} \left[ V_1^\star(s_1^k) - V_1^{\pi^k}(s_1^k) \right] &\le \sum_{h=1}^H \sum_{k \in \mathcal{I}_h^K} \left[ Q_h^{k-1}(s_h^k, a_h^k) - Q_h^\star(s_h^k, a_h^k) + (P_{h, s_h^k, a_h^k} - P_h^k)(V_{h+1}^\star - V_{h+1}^{\pi^k}) \right] \\
&\le \sum_{h=1}^H \sum_{k \in \mathcal{I}_h^K} \left[ 2 b_h^{k-1}(s_h^k, a_h^k) + (P_{h, s_h^k, a_h^k} - P_h^k)(V_{h+1}^\star - V_{h+1}^{\pi^k}) \right] \\
&= \underbrace{\sum_{h=1}^H \sum_{k \in \mathcal{I}_h^K} 2 b_h^{k-1}(s_h^k, a_h^k)}_{=: \xi_1} + \underbrace{\sum_{h=1}^H \sum_{k \in \mathcal{I}_h^K} (P_{h, s_h^k, a_h^k} - P_h^k)(V_{h+1}^\star - V_{h+1}^{\pi^k})}_{=: \xi_2},
\end{aligned}
$$

where the second inequality invokes Lemma 2. Therefore, it is sufficient to bound $\xi_1$ and $\xi_2$ separately, which we accomplish as follows.

- Regarding the term $\xi_2$, we first make the observation that $\left\{ (P_{h, s_h^k, a_h^k} - P_h^k)(V_{h+1}^\star - V_{h+1}^{\pi^k}) \right\}$ forms a martingale difference sequence, as $\pi_j^k$ is determined by $Q_j^{k-1}(s_j^k, a)$. Moreover, the sequence satisfies the trivial bound

$$
\left| (P_{h, s_h^k, a_h^k} - P_h^k)(V_{h+1}^\star - V_{h+1}^{\pi^k}) \right| \le H.
$$

These properties allow us to apply the celebrated Azuma-Hoeffding inequality (Azuma, 1967), which together with the trivial upper bound $\sum_{h=1}^H |\mathcal{I}_h^K| \le KH$ ensures that

$$
|\xi_2| = \left| \sum_{h=1}^H \sum_{k \in \mathcal{I}_h^K} (P_{h, s_h^k, a_h^k} - P_h^k)(V_{h+1}^\star - V_{h+1}^{\pi^k}) \right| \le H \sqrt{HK \log \frac{2}{\delta}}
\tag{31}
$$

with probability at least $1 - \delta$.

- Turning to the term $\xi_1$, we apply the Cauchy-Schwarz inequality to derive

$$
\xi_1 = \sum_{h=1}^H \sum_{k \in \mathcal{I}_h^K} 2\beta \sqrt{\varphi_h(s_h^k, a_h^k)^\top (\Lambda_h^{k-1})^{-1} \varphi_h(s_h^k, a_h^k)}
$$

$$\leq 2\beta\sqrt{KH}\sqrt{\sum_{h=1}^{H}\sum_{k\in\mathcal{I}_h^K}\varphi_h(s_h^k,a_h^k)^\top\left(\Lambda_h^{k-1}\right)^{-1}\varphi_h(s_h^k,a_h^k)}. \tag{32}$$

To further control the right-hand side of (32), we resort to Lemma 4 in Section C.3 — a result borrowed from (Abbasi-Yadkori et al., 2011) — which immediately leads to

$$\xi_1 \leq 2\beta\sqrt{KH}\cdot\sqrt{2Hd\log(KH)} = 2H\beta\sqrt{2dK\log(KH)}. \tag{33}$$

Putting everything together gives

$$\sum_{k\in\mathcal{I}_1^K}\left[V_1^\star(s_1^k)-V_1^{\pi^k}(s_1^k)\right] \leq 2H\beta\sqrt{2dK\log(KH)}+\sqrt{H^3K\log\frac{2}{\delta}} \leq 4c_\beta\sqrt{d^2H^6K\log^2\frac{KH}{\delta}}, \tag{34}$$

where the last inequality makes use of the definition $\beta := c_\beta\sqrt{dH^4\log\frac{KH}{\delta}}$.

**Step 3: bounding the number of state revisits.** To this end, we make the observation that $N = |\mathcal{I}_1^K|$ and $K = |\mathcal{I}_H^K|$ (see Lemma 1). With this in mind, we can bound the total number $K-N$ of state revisits as follows:

$$K-N = |\mathcal{I}_H^K|-|\mathcal{I}_1^K| = \sum_{h=1}^{H}\left(|\mathcal{I}_{h+1}^K|-|\mathcal{I}_h^K|\right) \leq \frac{4c_\beta^2 d^2H^5\log^2\frac{KH}{\delta}}{\Delta_{\mathsf{gap}}^2}. \tag{35}$$

Here, the above inequality is a consequence of the auxiliary lemma below, whose proof is provided in Section C.2.

**Lemma 3.** *Suppose that $KH \geq 2$. For all $1 \leq h < H$, the following condition*

$$\left|\mathcal{I}_{h+1}^K\setminus\mathcal{I}_h^K\right| \leq \frac{4c_\beta^2 d^2H^4\log^2\frac{KH}{\delta}}{\Delta_{\mathsf{gap}}^2} \tag{36}$$

*holds, where $c_\beta > 0$ is the pre-constant defined in* (16).

**Step 4: sample complexity analysis.** Recall that $T$ stands for the total number of samples collected, which clearly satisfies $T \geq K$. Consequently, the above results (35) and (34) taken collectively lead to

$$0 \leq K-N \leq \frac{4c_\beta^2 d^2H^5\log^2\frac{TH}{\delta}}{\Delta_{\mathsf{gap}}^2} \leq N \qquad\Longrightarrow\qquad N \leq K \leq 2N, \tag{37}$$

provided that $N \geq \frac{4c_\beta^2 d^2H^5\log^2\frac{HT}{\delta}}{\Delta_{\mathsf{gap}}^2}$. This together with the fact $T \leq KH$ implies that

$$T \leq KH \leq 2NH \qquad\Longrightarrow\qquad N \geq \frac{T}{2H}. \tag{38}$$

As a result, we can invoke (34) to obtain

$$\frac{1}{N}\sum_{n=1}^{N}\left(V_1^\star(s_1^{(n)})-V_1^{\pi^{(n)}}(s_1^{(n)})\right) = \frac{1}{N}\sum_{k\in\mathcal{I}_1^K}\left[V_1^\star(s_1^k)-V_1^{\pi^k}(s_1^k)\right] \leq 4c_\beta\sqrt{\frac{d^2H^6K\log^2\frac{HT}{\delta}}{N^2}}$$

$$\leq 8c_\beta\sqrt{\frac{d^2H^7\log^2\frac{HT}{\delta}}{T}}, \tag{39}$$

where the last relation arises from both (37) and (38). This concludes the proof.

## B.2 Analysis for regret over the paths (proof of (21))

**Proof of (21a).** Invoking the crude bound $V^\star(s_1^k) - V^{\pi^k}(s_1^k) \leq H$ leads to

$$
\begin{aligned}
\mathsf{Regret}_{\mathsf{path}}(K) &= \sum_{k \in \mathcal{I}_1^K} \left[ V_1^\star(s_1^k) - V_1^{\pi^k}(s_1^k) \right] + \sum_{k \notin \mathcal{I}_1^K} \left[ V_1^\star(s_1^k) - V_1^{\pi^k}(s_1^k) \right] \\
&\leq \sum_{k \in \mathcal{I}_1^K} \left[ V_1^\star(s_1^k) - V_1^{\pi^k}(s_1^k) \right] + H \big( K - |\mathcal{I}_1^K| \big) \\
&= \sum_{k \in \mathcal{I}_1^K} \left[ V_1^\star(s_1^k) - V_1^{\pi^k}(s_1^k) \right] + H(K - N) \\
&\leq 4 c_\beta \sqrt{d^2 H^6 K \log^2 \frac{HT}{\delta}} + \frac{4 c_\beta^2 d^2 H^6 \log^2 \frac{TH}{\delta}}{\Delta_{\mathsf{gap}}^2},
\end{aligned} \tag{40}
$$

where the penultimate line relies on (26), and the last relation holds due to (34) and (35).

**Proof of (21b) (logarithmic regret).** As it turns out, this logarithmic regret bound (w.r.t. $K$) can be established by combining our result with a result derived in Yang et al. (2021). To be precise, by defining $\Delta_h(s, a) := V_h^\star(s) - Q_h^\star(s, a)$, we make the following observation:

$$
\mathsf{Regret}_{\mathsf{path}}(K) = \sum_{k=1}^K \left( V_1^\star(s_1^k) - V_1^{\pi^k}(s_1^k) \right) = \sum_{k=1}^K \mathbb{E}\left[ \sum_{h=1}^H \Delta_h(s_h^k, a_h^k) \mid \pi^k, s_1^k \right],
$$

which has been derived in Yang et al. (2021, Equation (1)). Unconditioning gives

$$
\begin{aligned}
\mathbb{E}\big[\mathsf{Regret}_{\mathsf{path}}(K)\big] &= \mathbb{E}\left[ \sum_{k=1}^K \left( V_1^\star(s_1^k) - V_1^{\pi^k}(s_1^k) \right) \right] = \mathbb{E}\left[ \sum_{k=1}^K \sum_{h=1}^H \Delta_h(s_h^k, a_h^k) \right] = \mathbb{E}\left[ \sum_{h=1}^H \sum_{k=1}^K \Delta_h(s_h^k, a_h^k) \right] \\
&= \mathbb{E}\left[ \sum_{h=1}^H \sum_{k \in \mathcal{I}_h^K} \Delta_h(s_h^k, a_h^k) \right] + \mathbb{E}\left[ \sum_{h=1}^H \sum_{k \notin \mathcal{I}_h^K} \Delta_h(s_h^k, a_h^k) \right].
\end{aligned} \tag{41}
$$

In addition, we make note of the fact that: with probability at least $1 - \delta$, one has

$$
\Delta_h(s_h^k, a_h^k) = 0 \qquad \text{for all} \quad k \in \mathcal{I}_h^K,
$$

which follows immediately from the update rule of Algorithm 2 (cf. line 14) and Lemma 2. This taken collectively with the trivial bound $\Delta_h(s_h^k, a_h^k) \leq H$ gives

$$
\mathbb{E}\left[ \sum_{h=1}^H \sum_{k \in \mathcal{I}_h^K} \Delta_h(s_h^k, a_h^k) \right] \leq (1 - \delta) \cdot 0 + \delta \cdot \sum_{h=1}^H \sum_{k \in \mathcal{I}_h^K} H \leq H^2 K \delta.
$$

Substitution into (41) yields

$$
\begin{aligned}
\mathbb{E}\big[\mathsf{Regret}_{\mathsf{path}}(K)\big] &\leq H^2 K \delta + \sum_{h=1}^H \sum_{k \notin \mathcal{I}_h^K} \mathbb{E}\big[\Delta_h(s_h^k, a_h^k)\big] \leq H^2 K \delta + \sum_{h=1}^H \sum_{k \notin \mathcal{I}_h^K} H \\
&\leq H^2 K \delta + H^2 (K - N) \leq H^2 K \delta + \frac{4 c_\beta^2 d^2 H^7 \log^2 \frac{TH}{\delta}}{\Delta_{\mathsf{gap}}^2}.
\end{aligned}
$$

Here, the second inequality follows from the trivial upper bound $\Delta_h(s_h^k, a_h^k) \leq H$, the third inequality holds true since $K - |\mathcal{I}_h^K| \leq K - |\mathcal{I}_H^K| = K - N$ (see Lemma 1), whereas the last inequality is valid due to (35). Taking $\delta = 1/K$ and recalling that $T \leq KH$, we arrive at the advertised logarithmic regret bound:

$$
\mathbb{E}\big[\mathsf{Regret}_{\mathsf{path}}(K)\big] \leq \frac{17 c_\beta^2 d^2 H^7 \log^2(KH)}{\Delta_{\mathsf{gap}}^2}.
$$

# C  Proof of technical lemmas

## C.1  Proof of Lemma 2

**Step 1: decomposition of $\theta_h^k - \theta_h^\star$.**    To begin with, recalling the update rule (15b), we have the following decomposition

$$\theta_h^k - \theta_h^\star = (\Lambda_h^k)^{-1} \left\{ \sum_{i \in \mathcal{I}_h^k} \varphi_h(s_h^i, a_h^i) \left[ r_h(s_h^i, a_h^i) + \langle \varphi_{h+1}(s_{h+1}^i, a_{h+1}^i), \theta_{h+1}^k \rangle \right] - \Lambda_h^k \theta_h^\star \right\}$$

$$= (\Lambda_h^k)^{-1} \left\{ \sum_{i \in \mathcal{I}_h^k} \varphi_h(s_h^i, a_h^i) \left[ \langle \varphi_{h+1}(s_{h+1}^i, a_{h+1}^i), \theta_{h+1}^k \rangle - P_{h, s_h^i, a_h^i} V_{h+1}^\star \right] - \theta_h^\star \right\}. \quad (42)$$

To see why the second identity holds, note that from the definition of $\Lambda_h^k$ (cf. (15a)) we have

$$\Lambda_h^k \theta_h^\star = \sum_{i \in \mathcal{I}_h^k} \varphi_h(s_h^i, a_h^i) \big( \varphi_h(s_h^i, a_h^i) \big)^\top \theta_h^\star + \theta_h^\star$$

$$= \sum_{i \in \mathcal{I}_h^k} \varphi_h(s_h^i, a_h^i) Q_h^\star(s_h^i, a_h^i) + \theta_h^\star$$

$$= \sum_{i \in \mathcal{I}_h^k} \varphi_h(s_h^i, a_h^i) \left[ r_h(s_h^i, a_h^i) + P_{h, s_h^i, a_h^i} V_{h+1}^\star \right] + \theta_h^\star,$$

where the second and the third identities invoke the linear realizability assumption of $Q_h^\star$ and the Bellman equation, respectively.

As a result of (42), to control the difference $\theta_h^k - \theta_h^\star$, it is sufficient to bound $\langle \varphi_{h+1}(s_{h+1}^i, a_{h+1}^i), \theta_{h+1}^k \rangle - P_{h, s_h^i, a_h^i} V_{h+1}^\star$. Towards this, we start with the following decomposition

$$\langle \varphi_{h+1}(s_{h+1}^i, a_{h+1}^i), \theta_{h+1}^k \rangle - P_{h, s_h^i, a_h^i} V_{h+1}^\star = \langle \varphi_{h+1}(s_{h+1}^i, a_{h+1}^i), \theta_{h+1}^k \rangle - Q_{h+1}^\star(s_{h+1}^i, a_{h+1}^i)$$

$$+ Q_{h+1}^\star(s_{h+1}^i, a_{h+1}^i) - V_{h+1}^\star(s_{h+1}^i) + V_{h+1}^\star(s_{h+1}^i) - P_{h, s_h^i, a_h^i} V_{h+1}^\star.$$

For notational simplicity, let us define

$$\varepsilon_h^k := \left[ \langle \varphi_h(s_h^i, a_h^i), \theta_h^k \rangle - Q_h^\star(s_h^i, a_h^i) \right]_{i \in \mathcal{I}_h^k} \qquad \in \mathbb{R}^{|\mathcal{I}_h^k|}, \quad (43a)$$

$$\delta_h^k := \left[ Q_{h+1}^\star(s_{h+1}^i, a_{h+1}^i) - V_{h+1}^\star(s_{h+1}^i) \right]_{i \in \mathcal{I}_h^k} \qquad \in \mathbb{R}^{|\mathcal{I}_h^k|}, \quad (43b)$$

$$\xi_h^k := \left[ V_{h+1}^\star(s_{h+1}^i) - P_{h, s_h^i, a_h^i} V_{h+1}^\star \right]_{i \in \mathcal{I}_h^k} \qquad \in \mathbb{R}^{|\mathcal{I}_h^k|}, \quad (43c)$$

$$\Phi_h^k := \left[ \varphi_h(s_h^i, a_h^i) \right]_{i \in \mathcal{I}_h^k} \qquad \in \mathbb{R}^{d \times |\mathcal{I}_h^k|}. \quad (43d)$$

Here and throughout, for any $z = [z_i]_{1 \leq i \leq K}$, the vector $[z_i]_{i \in \mathcal{I}_h^k}$ denotes a subvector of $z$ formed by the entries with indices coming from $\mathcal{I}_h^k$; for any set of vectors $w_1, \cdots, w_K$, the matrix $[w_i]_{i \in \mathcal{I}_h^k}$ represents a submatrix of $[w_1, \cdots, w_K]$ whose columns are formed by the vectors with indices coming from $\mathcal{I}_h^k$. Armed with this set of notation, $\theta_h^k - \theta_h^\star$ can be succinctly expressed as

$$\theta_h^k - \theta_h^\star = (\Lambda_h^k)^{-1} \left\{ \Phi_h^k \left( \left[ \varepsilon_{h+1}^k \right]_{i \in \mathcal{I}_h^k} + \delta_h^k + \xi_h^k \right) - \theta_h^\star \right\}, \quad (44)$$

where we further define

$$\left[ \varepsilon_{h+1}^k \right]_{i \in \mathcal{I}_h^k} := \left[ \langle \varphi_{h+1}(s_{h+1}^i, a_{h+1}^i), \theta_{h+1}^k \rangle - Q_{h+1}^\star(s_{h+1}^i, a_{h+1}^i) \right]_{i \in \mathcal{I}_{h+1}^k \cap \mathcal{I}_h^k};$$

in other words, we consider the vector $\varepsilon_{h+1}^k$ when restricted to the index set $\mathcal{I}_{h+1}^k \cap \mathcal{I}_h^k$. Recognizing that $\mathcal{I}_h^k \subseteq \mathcal{I}_{h+1}^k$ (see Lemma 1), we can also simply write

$$\left[ \varepsilon_{h+1}^k \right]_{i \in \mathcal{I}_h^k} = \left[ \langle \varphi_{h+1}(s_{h+1}^i, a_{h+1}^i), \theta_{h+1}^k \rangle - Q_{h+1}^\star(s_{h+1}^i, a_{h+1}^i) \right]_{i \in \mathcal{I}_h^k}.$$

**Step 2: decomposition of** $Q_h^k(s,a) - Q_h^\star(s,a)$**.** We now employ the above decomposition of $\theta_h^k - \theta_h^\star$ to help control $Q_h^k(s,a) - Q_h^\star(s,a)$ — the target quantity of Lemma 2. By virtue of the relation (44), our estimate $\langle \varphi_h(s,a), \theta_h^k \rangle$ of the linear representation $Q_h^\star(s,a) = \langle \varphi_h(s,a), \theta_h^\star \rangle$ satisfies

$$
\begin{aligned}
\left| \langle \varphi_h(s,a), \theta_h^k \rangle - Q_h^\star(s,a) \right| &= \left| \langle \varphi_h(s,a), \theta_h^k - \theta_h^\star \rangle \right| \\
&= \left| \varphi_h(s,a)^\top \left( \Lambda_h^k \right)^{-1} \left\{ \Phi_h^k \left( \left[ \varepsilon_{h+1}^k \right]_{i \in \mathcal{I}_h^k} + \delta_h^k + \xi_h^k \right) - \theta_h^\star \right\} \right| \\
&\leq \left\| \left( \Lambda_h^k \right)^{-1/2} \varphi_h(s,a) \right\|_2 \cdot \\
&\quad \left( \left\| \left( \Lambda_h^k \right)^{-1/2} \Phi_h^k \right\| \left\{ \left\| \varepsilon_{h+1}^k \right\|_2 + \left\| \delta_h^k \right\|_2 \right\} + \left\| \left( \Lambda_h^k \right)^{-1/2} \Phi_h^k \xi_h^k \right\|_2 + \left\| \left( \Lambda_h^k \right)^{-1/2} \right\| \left\| \theta_h^\star \right\|_2 \right),
\end{aligned}
$$

where $\|M\|$ denotes the spectral norm of a matrix $M$. Here, the last inequality follows from the Cauchy-Schwarz inequality and the triangle inequality. Now, from the definition

$$
\Lambda_h^k = \sum_{i \in \mathcal{I}_h^k} \varphi_h(s_h^i, a_h^i) \varphi_h(s_h^i, a_h^i)^\top + I = \Phi_h^k (\Phi_h^k)^\top + I, \tag{45}
$$

it is easily seen that $\left\| \left( \Lambda_h^k \right)^{-1/2} \right\| \leq 1$ and $\left\| \left( \Lambda_h^k \right)^{-1/2} \Phi_h^k \right\| \leq 1$. Consequently, it is guaranteed that

$$
\begin{aligned}
&\left| \langle \varphi_h(s,a), \theta_h^k \rangle - Q_h^\star(s,a) \right| \\
&\leq \left( \left\| \varepsilon_{h+1}^k \right\|_2 + \left\| \delta_h^k \right\|_2 + \left\| \left( \Lambda_h^k \right)^{-1/2} \Phi_h^k \xi_h^k \right\|_2 + \left\| \theta_h^\star \right\|_2 \right) \cdot \left\| \left( \Lambda_h^k \right)^{-1/2} \varphi_h(s,a) \right\|_2. \tag{46}
\end{aligned}
$$

In the sequel, we seek to establish, by induction, that

$$
\left| \langle \varphi_h(s,a), \theta_h^k \rangle - Q_h^\star(s,a) \right| \leq b_h^k(s,a). \tag{47}
$$

If this condition were true, then combining this with the definition (see (15d))

$$
Q_h^k(s,a) = \min \left\{ \langle \varphi_h(s,a), \theta_h^k \rangle + b_h^k(s,a), \, H \right\}
$$

and the constraint $Q_h^\star(s,a) \leq H$ would immediately lead to

$$
Q_h^k(s,a) - Q_h^\star(s,a) \leq \left| \langle \varphi_h(s,a), \theta_h^k \rangle + b_h^k(s,a) - Q_h^\star(s,a) \right| \leq 2 b_h^k(s,a),
$$

$$
Q_h^k(s,a) - Q_h^\star(s,a) \geq \begin{cases} 0, & \text{if } Q_h^k(s,a) \geq H, \\ b_h^k(s,a) - \left| \langle \varphi_h(s,a), \theta_h^k \rangle - Q_h^\star(s,a) \right| \geq 0, & \text{else,} \end{cases}
$$

as claimed in the inequality (28) of this lemma. Consequently, everything boils down to establishing (47), which forms the main content of the next setp.

**Step 3: proof of the inequality** (47)**.** The proof of this inequality proceeds by bounding each term in the relation (46).

To start with, we establish an upper bound on $\left\| \left( \Lambda_h^k \right)^{-1/2} \Phi_h^k \xi_h^k \right\|_2$ as required in our advertised inequality (47); that is, with probability at least $1 - \delta$,

$$
\left\| \left( \Lambda_j^k \right)^{-1/2} \Phi_j^k \xi_j^k \right\|_2 \leq \sqrt{H^2 \left( d \log(KH) + 2 \log \frac{KH}{\delta} \right)} \leq 2 \sqrt{H^2 d \log \frac{KH}{\delta}} \tag{48}
$$

holds simultaneously for all $1 \leq j \leq H$ and $1 \leq k \leq K$. Towards this end, let us define

$$
X_i := V_{j+1}^\star(s_{j+1}^i) - P_{j, s_j^i, a_j^i} V_{j+1}^\star. \tag{49}
$$

It is easily seen that $\{X_i\}_{i \in \mathcal{I}_j^K}$ forms a martingale sequence. In addition, we have the trivial upper bound $|X_i| \leq H$. Therefore, applying the concentration inequality for self-normalized processes (see Lemma 4 in Section C.3), we can deduce that

$$
\left\| \left( \Lambda_j^k \right)^{-1/2} \Phi_j^k \xi_j^k \right\|_2 = \left\| \left( \Lambda_j^k \right)^{-1/2} \sum_{i \in \mathcal{I}_j^k} \varphi_j(s_j^i, a_j^i) X_i \right\|_2 \leq \sqrt{H^2 \log \left( \frac{\det \left( \Lambda_j^k \right)}{\det \left( \Lambda_j^0 \right) \delta^2} \right)}
$$

$$\leq \sqrt{H^2 \left( d \log(KH) + 2 \log \frac{1}{\delta} \right)} \tag{50}$$

holds with probability at least $1 - \delta$. Here, the first inequality comes from Lemma 5 in Section C.3, whereas the second inequality is a consequence of Lemma 4 in Section C.3. Taking the union bound over $j = 1, \ldots, H$ and $k = 1, \ldots, K$ yields the required relation (48).

Armed with the above inequality, we can move on to establish the inequality (47) by induction, working backwards. First, we observe that when $h = H + 1$, the inequality (47) holds true trivially (due to the initialization $\theta_{H+1}^k = 0$ and the fact $Q_{H+1}^\star = 0$). Next, let us assume that the claim holds for $h + 1, \ldots, H + 1$, and show that the claim continues to hold for step $h$. To this end, it is sufficient to bound the terms in (46) separately.

- **The term** $\|\delta_j^k\|_2$. From the induction hypothesis, the inequality (47) holds for $h + 1, \ldots, H + 1$. For any $j$ obeying $h \leq j \leq H$ and any $i \in \mathcal{I}_j^k$, this in turn guarantees that

$$V_{j+1}^\star(s_{j+1}^i) = Q_{j+1}^\star\big(s_{j+1}^i, \pi^\star(s_{j+1}^i)\big) \leq Q_{j+1}^{i-1}\big(s_{j+1}^i, \pi^\star(s_{j+1}^i)\big) \leq Q_{j+1}^{i-1}(s_{j+1}^i, a_{j+1}^i)$$
$$\leq \big\langle \varphi_h(s_{j+1}^i, a_{j+1}^i), \theta_{j+1}^{i-1} \big\rangle + b_{j+1}^{i-1}(s_{j+1}^i, a_{j+1}^i).$$

Here, the second inequality follows since $a_{j+1}^i$ is chosen to be an action maximizing $Q_{j+1}^{i-1}(s_{j+1}^i, \cdot)$. In the meantime, the induction hypothesis (47) for $h + 1, \ldots, H + 1$ also implies

$$Q_{j+1}^\star(s_{j+1}^i, a_{j+1}^i) \geq \big\langle \varphi_h(s_{j+1}^i, a_{j+1}^i), \theta_{j+1}^{i-1} \big\rangle - b_{j+1}^{i-1}(s_{j+1}^i, a_{j+1}^i)$$

for all $j$ obeying $h \leq j \leq H$. Taken collectively, the above two inequalities demonstrate that

$$0 \leq V_{j+1}^\star(s_{j+1}^i) - Q_{j+1}^\star(s_{j+1}^i, a_{j+1}^i) \leq 2b_{j+1}^{i-1}(s_{j+1}^i, a_{j+1}^i), \qquad i \in \mathcal{I}_j^k,$$

where the first inequality holds trivially since $V_{j+1}^\star(s) = \max_a Q_{j+1}^\star(s, a)$. Then, given that $i \in \mathcal{I}_j^k$, one necessarily has $b_{j+1}^{i-1}(s_{j+1}^i, a_{j+1}^i) < \Delta_{\mathsf{gap}}/2$, which combined with the sub-optimality gap assumption (9) implies that $a_{j+1}^i$ cannot be a sub-optimal action in state $s_{j+1}^i$. Consequently, we reach

$$V_{j+1}^\star(s_{j+1}^i) - Q_{j+1}^\star(s_{j+1}^i, a_{j+1}^i) = 0 \qquad \text{for all } i \in \mathcal{I}_j^k,$$

and as a result,

$$\delta_j^k = 0. \tag{51}$$

- **The term** $\|\varepsilon_{h+1}^k\|_2$. Recall that the decomposition (44) together with the assumption $Q_h^\star(s, a) = \langle \varphi_h(s, a), \theta_h^\star \rangle$ allows us to write $\varepsilon_h^k$ (cf. (43a)) as follows

$$\varepsilon_h^k = \big(\Phi_h^k\big)^\top \big(\theta_h^k - \theta_h^\star\big) = \big(\Phi_h^k\big)^\top \big(\Lambda_h^k\big)^{-1} \Big\{ \Phi_h^k\big( \big[\varepsilon_{h+1}^k\big]_{i \in \mathcal{I}_h^k} + \delta_h^k + \xi_h^k \big) - \theta_h^\star \Big\}.$$

Combining this with the basic properties $\|(\Phi_h^k)^\top (\Lambda_h^k)^{-1/2}\| \leq 1$, $\|(\Lambda_h^k)^{-1/2}\| \leq 1$, and $\mathcal{I}_h^K \subseteq \mathcal{I}_{h+1}^K$ yields

$$\|\varepsilon_h^k\|_2 = \left\| \big(\Phi_h^k\big)^\top \big(\Lambda_h^k\big)^{-1} \Big\{ \Phi_h^k\big( \big[\varepsilon_{h+1}^k\big]_{i \in \mathcal{I}_h^k} + \delta_h^k + \xi_h^k \big) - \theta_h^\star \Big\} \right\|_2$$
$$\leq \|\varepsilon_{h+1}^k\|_2 + \|\delta_h^k\|_2 + \left\| \big(\Lambda_h^k\big)^{-1/2} \Phi_h^k \xi_h^k \right\|_2 + \|\theta_h^\star\|_2. \tag{52}$$

Applying this inequality recursively leads to

$$\|\varepsilon_h^k\|_2 \leq \|\varepsilon_{H+1}^k\|_2 + \sum_{h \leq j \leq H} \left[ \|\delta_j^k\|_2 + \left\| \big(\Lambda_h^k\big)^{-1/2} \Phi_j^k \xi_j^k \right\|_2 + \|\theta_j^\star\|_2 \right]$$
$$\leq 4\sqrt{dH^4 \log \frac{KH}{\delta}}, \tag{53}$$

where the last inequality holds by putting together the property $\varepsilon_{H+1}^k = 0$, the inequalities (48) and (51), and the assumption that $\|\theta_j^\star\|_2 \leq 2H\sqrt{d}$ (see (8)).

Combining the inequalities (48), (51), (53) with the relation (46), we arrive at

$$\left|\langle\varphi_h(s,a),\theta_h^k\rangle - Q_h^\star(s,a)\right| \le \left(4\sqrt{dH^4\log\frac{KH}{\delta}} + 2\sqrt{H^2d\log\frac{KH}{\delta}} + 2H\sqrt{d}\right)\left\|(\Lambda_h^k)^{-1/2}\varphi_h(s,a)\right\|_2$$

$$\le c_\beta\sqrt{dH^4\log\frac{KH}{\delta}}\left\|(\Lambda_h^k)^{-1/2}\varphi_h(s,a)\right\|_2 = b_h^k(s,a),$$

provided that $c_\beta \ge 8$. This completes the induction step of (47), thus concluding the proof of Lemma 2.

## C.2 Proof of Lemma 3

Consider any $1 \le h < H$. In view of the definition (15c) of the UCB bonus $b_h^k$, one obtains

$$\sum_{k:k+1\in\mathcal{I}_{h+1}^K}\left[b_{h+1}^{k-1}(s_{h+1}^k,a_{h+1}^k)\right]^2 \le \sum_{k:k+1\in\mathcal{I}_{h+1}^K}c_\beta^2 dH^4\left[\varphi_{h+1}(s_{h+1}^k,a_{h+1}^k)\right]^\top\left(\Lambda_{h+1}^{k-1}\right)^{-1}\varphi_{h+1}(s_{h+1}^k,a_{h+1}^k)\log\frac{KH}{\delta}$$

$$\le c_\beta^2 d^2 H^4\log\frac{KH}{\delta}\log\frac{K+d}{d} \le c_\beta^2 d^2 H^4\log^2\frac{KH}{\delta} =: B, \tag{54}$$

where the second inequality comes from a standard result stated in Lemma 4 of Section C.3, and the last inequality is valid as long as $KH \ge 2$.

As a direct consequence of the upper bound (54), there are no more than $\frac{B}{(\Delta_{\mathsf{gap}}/2)^2}$ elements in the set $\mathcal{I}_{h+1}^K$ obeying $b_{h+1}^{k-1}(s_{h+1}^k,a_{h+1}^k) \ge \Delta_{\mathsf{gap}}/2$. This combined with the update rule in Line 14 of Algorithm 2 immediately yields

$$\left|\mathcal{I}_{h+1}^K \setminus \mathcal{I}_h^K\right| \le \frac{B}{(\Delta_{\mathsf{gap}}/2)^2} = \frac{4c_\beta^2 d^2 H^4\log^2\frac{KH}{\delta}}{\Delta_{\mathsf{gap}}^2},$$

since, by construction, any $k$ with $k+1$ belonging to $\mathcal{I}_{h+1}^K \setminus \mathcal{I}_h^K$ must violate the condition $b_{h+1}^{k-1}(s_{h+1}^k,a_{h+1}^k) < \Delta_{\mathsf{gap}}/2$. This completes the proof.

## C.3 Auxiliary lemmas

In this section, we provide a couple of auxiliary lemmas that have been frequently invoked in the literature of linear bandits and linear MDPs. The first result is concerned with the interplay between the feature map and the (regularized) covariance matrix $\Lambda_h^k$.

**Lemma 4** (Abbasi-Yadkori et al. (2011)). *Under the assumption* (8) *and the definition* (15a), *the following relationship holds true*

$$\sum_{i\in\mathcal{I}_h^k}\varphi_h(s_h^i,a_h^i)^\top\left(\Lambda_h^{i-1}\right)^{-1}\varphi_h(s_h^i,a_h^i) \le 2\log\left(\frac{\det\left(\Lambda_h^k\right)}{\det\left(\Lambda_h^0\right)}\right) \le 2d\log\left(\frac{k}{d}+1\right). \tag{55}$$

*Proof.* The first inequality is an immediate consequence of Abbasi-Yadkori et al. (2011) or Jin et al. (2020, Lemma D.2). Regarding the second inequality, let $\lambda_i \ge 1$ be the $i$-th largest eigenvalue of the positive-semidefinite matrix $\Lambda_h^k$. From the AM-GM inequality, it is seen that

$$\det\left(\Lambda_h^k\right) = \prod_{i=1}^d \lambda_i \le \left(\frac{\sum_{i=1}^d \lambda_i}{d}\right)^d \le \left(\frac{k+d}{d}\right)^d, \tag{56}$$

where the last inequality arises since (in view of the assumption (8))

$$\sum_{i=1}^d \lambda_i = \mathsf{Tr}\left(\Lambda_h^k\right) = d + \sum_{i\in\mathcal{I}_h^k}\left\|\varphi_h(s_h^i,a_h^i)\right\|_2^2 \le d + k.$$

$\square$

The second result delivers a concentration inequality for the so-called self-normalized processes.

**Lemma 5** (Abbasi-Yadkori et al. (2011)). *Assume that $\{X_t \in \mathbb{R}\}_{t=1}^{\infty}$ is a martingale w.r.t. the filtration $\{\mathcal{F}_t\}_{t=0}^{\infty}$ obeying*

$$\mathbb{E}\left[X_t \,|\, \mathcal{F}_{t-1}\right] = 0, \qquad and \qquad \mathbb{E}\left[e^{\lambda X_t} \,|\, \mathcal{F}_{t-1}\right] \leq e^{\frac{\lambda^2 \sigma^2}{2}}, \ \forall \, \lambda \in \mathbb{R}.$$

*In addition, suppose that $\varphi_t \in \mathbb{R}^d$ is a random vector over $\mathcal{F}_{t-1}$, and define $\Lambda_t = \Lambda_0 + \sum_{i=1}^{t} \varphi_i \varphi_i^{\top} \in \mathbb{R}^{d \times d}$. Then with probability at least $1 - \delta$, it follows that*

$$\left\| (\Lambda_t)^{-1/2} \sum_{i=1}^{t} \varphi_i X_i \right\|_2^2 \leq \sigma^2 \log\left( \frac{\det(\Lambda_t)}{\det(\Lambda_0)\delta^2} \right) \qquad for \ all \ t \geq 0.$$