# OpenReview forum: "Sample-Efficient Reinforcement Learning Is Feasible for Linearly Realizable MDPs with Limited Revisiting"
_NeurIPS.cc/2021/Conference — NeurIPS 2021 Poster_

### Official Review · Reviewer_fsA9 · 2021-07-09

**Rating:** 6
**Confidence:** 4

**Summary:**

This paper introduces a new problem setting of RL with state resetting. It then proves a polynomial (in ) sample complexity bound under linear approximation with assumptions of (1) $Q^*$ realizability and (2) suboptimality gap $\Delta $ bounded away from 0. This setting and bound helps to reconcile recent results under assumptions (1) and (2) of exponential lower bounds in the online setting with polynomial upper bounds in the generative model setting.

**Limitations And Societal Impact:**

The paper does a poor job discussing the limitations of the approach. This seems to largely be due to space constraints as the paper didn't even have space to really discuss the main technical contributions. However, the paper could benefit from a better discussion of limitations and future directions that could build on the idea state resetting.

**Main Review:**

### Strengths:

1. The result closes a gap in the literature. Namely, it helps to explain what sort of sampling procedure may be required to achieve polynomial complexity under $ Q^* $ realizability without a full generative model. This can help people working in this subfield to understand more precisely where the hardness of the online problem is coming from and could potentially inspire future work that leverages this insight.
2. The algorithm technique is interesting and proof is clean. I especially liked the way that the index set $\mathcal{I}$ is used throughout to very elegantly keep track of the resetting. The technical details of the proof are clearly presented in the appendix.
3. The presentation of related work is clear and comprehensive. The paper clearly explains the state of the literature and the existence of a gap in the realizability plus gap setting between online and generative model sampling procedures. The paper clearly explains the background on linearity assumptions in RL.

---

### Weaknesses:

1. The paper has a few issues with clarity with respect to the novel contribution. While the paper clearly presents related work and setup/background, it ends up spending so much space on it that the presentation of the novel results is rushed and often unclear. I think the paper would benefit from more exposition about the novel components described below.

   a) The sample complexity definition is non-standard. This is necessary because the standard definitions do not quite make sense in the novel setting. My understanding is that the regret definition captures regret of each "episode", but with the state resetting this is not sufficient because it does not measure regret suffered after each reset on the new "paths". So, the paper must consider the sample complexity that accounts for the samples collected on the reset paths. However instead of considering a standard notion of sample complexity as the number of samples until we arrive at an epsilon optimal policy, the paper considers sample complexity until the average regret is below epsilon. This definition is a sort of mash-up between regret and sample complexity that at least to me does not fit very obviously into the literature. More discussion and clarity around this definition would be helpful.

   b) The paper does not provide an adequate description of the resetting strategy. After staring at the algorithm for a while it seems to me that the idea is to continue resetting to step $ h $ until we have enough certainty about  $ s_h, a_h $ that we can make an update with uncertainty that is smaller than the gap $ \Delta$. Intuitively this requires very many resets, but as the proof shows it is polynomially bounded. This was never really explained in the text and definitely caused confusion for me when reading it for the first time. This seems to be the main algorithmic contribution, so it should be emphasized more.

   c) The proof is entirely relegated to the appendix. It would be helpful to at least give some high level ideas about the structure of the proof in the main text. I understand this was likely cut due to space constraints, but since it seems to be the main contribution of the paper, it would be nice to see more discussion of it in the paper itself.

2. I am somewhat wary of the significance of the result for a few reasons:

   a) It is not clear that the linearly realizable resetting setting is very useful beyond being a theoretical curiosity. The one example provided in the paper is that video games can be reset to previously seen states. I agree that with access to a simulator, state resetting is possible. But it also seems that with access to a simulator, more aggressive sorts of access to the generative model are also possible so it is not clear why one would need to constrain themselves to previously seen states. And on the other hand, in the real world resetting is clearly not possible.

   b) The paper only considers one very specific set of assumptions: realizability plus gap. This is not to say that it is not a novel contribution, only that the impact of results in this setting does not have clear implications for practice or beyond this specific setup. It would be useful to examine the effect of access to state resetting under slight variants of this setting as well. For example: Does it help without the gap? Does it help with $ Q^\pi$ realizability? Answering questions like these would boost the significance.


---

### Recommendation:

Weak accept.

I think the paper has a nice contribution that closes a gap in the literature with a technically elegant solution. However, I am wary about some issues with clarity and significance so I recommend a weak accept.

---

### Typos/minor comments/questions:

1. The computational complexity analysis seems wrong. At each $ k $, the computation in (15b) needs to take k inner products to compute the regression target $\langle \phi_i, \theta_{h+1}^k\rangle$. I think this cannot be pre-computed since the $ \theta_{h+1}^k$ is potentially different at each iteration. This means that the computation time should be $ O(d^2|\mathcal{A}| T^2)$.
2. Algorithm 3, line 9, the subscript should be $j$ instead of $h$
3. Equation 23 the superscripts should be $ K $ instead of $ k $



**Time Spent Reviewing:**

6

---

> ### Author Response · Authors · 2021-08-10
> **Thank you for the careful reading of the paper and insightful and valuable feedbacks! Below is a point-to-point response.**
>
> ## Regret bounds and sample complexity.
>
> Thanks for raising this issue. We will rephrase our main results in our paper by, for example,  including the usual full regret bound and adding expanded discussions about its connection to sample complexity to avoid confusion. As some quick remarks (which we will incorporate into the final version), Theorem 1 combined with some standard argument implies a PAC-based sample complexity with the same order as (17).
> In terms of a full regret bound, combining (34), (36), and the basic bound $V^{\star}(s_1^k) - V^{\pi^k}(s_1^k) \le H$ leads to
> $$ \mathsf{Regret} \le 4c_{\beta}\sqrt{d^2H^6K\log^2\frac{HT}{\delta}} + \frac{4c_{\beta}^2d^2H^6\log^2 \frac{TH}{\delta}}{\Delta_{\mathsf{gap}}^2}.$$
> We will add these detailed explanations in the paper to reconcile our work with related literature.
>
>
> ## Description of the resetting strategy.
> Thanks for your suggestion. In the final version, we will add more detailed explanations about the resetting strategy, and make sure to emphasize earlier that the number of resets can be well-controlled.
>
> ## Proof.
> Thanks for this suggestion. We will add a skeleton of the proof with intuitions in the main body in the final version.
>
> ## Resetting setting.
> Thanks for raising this issue. In addition to video games, state revisiting has also been considered in Monte Carlo Tree Search (MCTS), which assumes that the learner can go back to father nodes (i.e., previous states), which has broad real-world applications such as AlphaGo, AlphaZero and so on. Our revisiting framework at a high level is also close in nature to some prior work [Weisz et al. 2021a] (which we shall elaborate on in the revision). We will add more background discussions and motivating applications about the state revisiting setting.
>
> On the other hand, the generative model for linearly realizable MDPs assumes that the learner has access to a set of states, such that any feature can be expressed by the features corresponding to these states. This is substantially more restrictive than our setting, as our framework does not allow resetting to arbitrary states. We will add more explanations to make it more clear in the final version.
>
>
> ## Linear $Q^{\pi}$.
>
> Thanks for the great question, which we will discuss in more detail in the revised paper.
> For example, one possible direction that can be similarly dealt with is to replace the sub-optimality gap assumption with linear realizability of $Q^{\pi}$ plus additional low-dimension structure for $V^{\pi}$ for any near-optimal policy.
> In order to see this, by replacing $\theta_h^{\star}$ with $\theta_h^{\pi}$, the analysis in the proof of Lemma 2 still holds true for the near-optimal policy $\pi$ except that (43) should be considered for $V_{j+1}^{\pi}$ instead of $V_{j+1}^{\star}$. Hence, the additional low-dimension structure for $V^{\pi}$ is employed to ensure we can make this change. Also, replacing $\pi^{\star}$ with $\pi$ in the remaining parts makes the regret bound (36) hold w.r.t. $V_1^{\pi}$. Then, taking the sub-optimality of $\pi$ into account gives the desired result.
>
> Once again, this is an example of new settings implied by our current theoretical framework, and we will point out more alternatives and other implications of our theory in the revised paper.
>
>
> ## Computational complexity.
> Thanks for the question. Notice that the regression target $\sum_i \langle \varphi_i, \theta^k \rangle = \langle \sum_i \varphi_i, \theta^k \rangle$, and the term $\sum_i \varphi_i$ can be computed inductively, which leads to a computational complexity of $O(d^2|\mathcal{A}|T\log T)$. We will add detailed discussions about computational complexity in the revised paper.
>
> ## Typos.
> Thanks for pointing out the typos, which will be fixed in the final paper.

---

### Official Review · Reviewer_SUw4 · 2021-07-16

**Rating:** 6
**Confidence:** 3

**Summary:**

Recent work in RL has been seeking ways that RL can draw the kinds of
generalization that we see in supervised learning. Recently, a couple groups
obtained efficient RL algorithms by assuming that the reward and transition
probabilities are linear combinations of some set of basis functions. A
somewhat more appealing, weaker assumption would be for merely the Q-function
(state-action value function) to be linear. Unfortunately, recent works also
found hard examples of environments with linear Q-functions, that require
exponential sample complexity. Indeed, even with a simulator, there are
exponential lower bounds -- only recently have any kind of polynomial-time
guarantees, with a polynomial dependence on the smallest gap between the
value of any optimal action any any sub-optimal action.

This work proposes a new model in which linear Q-functions are learnable.
The model can be seen as a weakening of the simulator model: whereas the
simulator model allowed the learner to query any state-value pair to get a
sample of the reward and transition functions, this model merely allows the
learner to return to states that it previously visited on some trajectory.


**Limitations And Societal Impact:**

Yes, I think reasonably so.

**Main Review:**


I'm not aware of anything quite like the revisiting model being proposed
previously, and so that is novel, as is the kind of use that is made of it.
How applicable it is, is debatable. The paper claims that there are "several
realistic scenarios," but more or less only mentions save points in video
games as something beyond the simulator model.

The other angle one could take on this work is figuring out exactly how strong
of a model is necessary to learn MDPs with linear Q-functions. I'm not on
board with this take, though, because I don't think it's the case that we have a
ton of natural environments that are known to have linear Q-functions.
Rather, we were hoping that the linearity assumption would allow efficient
learners to exhibit generalization in a realistic learning model, and I think the
message of this line of work has been largely negative, that the linear
Q-function assumption just isn't adequate for this purpose. So rather than
pursuing linear Q-functions, we just have to try to find a different
assumption.

In summary, it seems like novel work in a novel but somewhat narrow model. I
am a little dubious about the long-term impact, since learning to play video
games was supposed to be a stepping stone to real-world problems, not an end
unto itself. It should probably be published somewhere, and NeurIPS has a
history of taking work on some oddball directions, and I don't think it would
*hurt* the conference to take this one. I wouldn't fight for it though.

**Time Spent Reviewing:**

3

---

> ### Author Response · Authors · 2021-08-10
> **Thank you for the careful reading of the paper and insightful and valuable feedbacks! Below is a point-to-point response.**
>
>
> ## Scenarios of state revisiting
>
> Thanks for raising this issue. Indeed, video game playing is one important that largely motivates the proposed model (in addition to the simulator setting). In addition to video games, state revisiting has also been considered in Monte Carlo Tree Search (MCTS), which assumes that the learner can go back to father nodes (i.e., previous states), which has found broad real-world applications such as AlphaGo, AlphaZero and so on. At a high level, our state revisiting framework is also close in nature to [Weisz et al. 2021a] (which we will elaborate on in the final version). We will add more backgrounds and motivating applications about the state revisiting setting in the revised paper.
>
>
> ## Linear Q-functions.
>
> Thanks for the comment. We agree with the reviewer that in practice, more complicated approximations (such as deep neural networks) are used  to tame the extremely high dimensionality (or even infinite dimensionality) of the state-action space in RL, with the premise that the optimal Q-function can be well-approximated by these low-complexity function classes. Due to their complex nature and intractability, linear approximations often serve as a prototype and a first step towards understanding more practical nonlinear function approximations.  While this work is limited towards addressing those more complicated models, we believe it provides some useful insights into the tractability of RL when the optimal Q-function enjoys low-dimensional structure (like linearly realizability). Compared to the linear MDP literature (where Q-functions under all policies are assumed to be linear), we believe this is one step towards relaxing the stringent linear assumptions imposed on both the transition kernel and the reward function.

---

### Official Review · Reviewer_w8yy · 2021-07-17

**Rating:** 7
**Confidence:** 3

**Summary:**

This paper studies the sample complexity of RL with the linear Q^* assumption. Previous work has shown that this problem can not be solved with polynomial samples neither in the generative setting nor in the online setting even with a large gap between the value of the best action and the second-best one. Here the authors show that the additional assumptions of a sampling procedure which allows the learner to backtrack to previously-seen states (together with a large gap) is sufficient for polynomial sample complexity (albeit with sample complexity scaling with the inverse gap). The algorithm is based on the “standard” LSVI-UCB template.

**Limitations And Societal Impact:**

As mentioned above there are a couple of technical issues which should be elaborated on/further developed and a couple of presentation issues (or related work issues) which should be corrected.

**Main Review:**

The results in this paper form a nice bridge between the online setting (where even a large gap does not suffice for sample-efficient learning) and the generative setting (where a gap condition can enable sample-efficient learning). The main sample complexity bound tells us that the full power of the generative setting is not needed and a “hybrid” interaction protocol can be sufficient. Since this result adds to the body of knowledge of sample efficient RL with linear function approximation, I am recommending acceptance, but with some suggestions/comments.

In terms of organization, the related works section should be moved out of the appendix and into the main paper. This can be merged with the discussion in section 1 and there are several other passages which repeat the same point (/are otherwise unnecessary) and can be cut (e.g. the “Sample Efficiency” paragraph of line 274 is not needed).

On the technical side: can the authors comment a bit more on the use of the gap condition? The “usual” application is that after poly(1/gap) number of samples at any state-action pair we have pi_k = pi^* for all future estimates. In Appendix C of Du et al. 2020a the estimates are performed at the anchor states which is then extrapolated over all features, giving pi_k = pi^star for that entire horizon and thus preventing the blowup of errors that comes with bootstrapping/LSVI methods. Is there a similar intuition here?

I am also wondering why the rewards are assumed to be deterministic, as I did not notice anywhere in the proofs where that was essential. Can the authors elaborate on this point?

The result is also framed in a somewhat abnormal way. It is neither a regret bound scaling with N nor a PAC bound on the risk, but rather a PAC bound for the average regret. Can a full regret bound be obtained?

In terms of related work: the “limited revisiting” framework introduced here is not new (despite the claims of Section 1.2). In fact the cited work of Weisz et. al 2021a introduced a similar framework.

Smaller comments on the writing:
- line 24: ‘’unprecedentedly enormous’’ <- why is it unprecedented?
- line 146: “The latter necessarily implies the former, while in contrast the former by no means implies the latter” <- that’s what being a weaker assumption means (stated in the previous sentence)



**Time Spent Reviewing:**

5

---

> ### Author Response · Authors · 2021-08-10
> **Thank you for the careful reading of the paper and insightful and valuable feedbacks! Below is a point-to-point response.**
>
> ## Organization
>
> Thanks for the suggestion. We will revise the paper accordingly: move related works into the main paper, and cut the unnecessary part as suggested.
>
> ## Technical use of the gap condition.
>
> Thanks for raising this point. To clarify, in the state revisiting setting, we cannot guarantee that $\pi_k = \pi^\star$ at any state-action pair for all future estimates after poly(1/gap) number of samples, which is different from the generative model setting in Du et al. 2020a. However, the gap condition ensures that there are at most poly(1/gap) number of samples such that $\pi_k \ne \pi^\star$, but the discrepency might happen at any time throughout the execution of the algorithm (rather than only happening at the beginning).
> In addition, the state revisiting setting helps the learner avoid these sub-optimal samples by resetting for at most poly(1/gap) times, which
> helps prevent the blowup of errors. We will add more explanations about this point in the final version.
>
> ## Deterministic rewards
>
> Thanks for raising this point. We totally agree that our analysis does not rely on deterministic rewards. It was assumed only for presentation simplicity. We will add clarifications as well as discussions about random rewards in the final version.
>
> ## Full regret bound
>
> Thanks for the suggestion. We will rephrase our main results and emphasize the full regret bound. As some quick remarks (which we will incorporate into the final version), Theorem 1 combined with some standard argument implies a PAC-based sample complexity with the same order as (17).  In terms of a full regret bound, combining (34), (36), and the basic bound $V^{\star}(s_1^k) - V^{\pi^k}(s_1^k) \le H$ leads to
> $$ \mathsf{Regret} \le 4c_{\beta}\sqrt{d^2H^6K\log^2\frac{HT}{\delta}} + \frac{4c_{\beta}^2d^2H^6\log^2 \frac{TH}{\delta}}{\Delta_{\mathsf{gap}}^2}. $$
> We will include detailed explanations of these points in the revision to avoid confusion.
>
> ## Related work.
>
> Thanks for brining this to our attention. In the final version, we will discuss [Weisz et al. 2021a] more accurately in the state revisiting framework, and revise our claims and discussions according throughout the paper.
>
> ## Comments on Writing.
>
> Thanks for the comments. We will fix these issues in the final paper.

---

> > ### Comment · Reviewer_w8yy · 2021-09-03
> > **Thanks for the reply**
> >
> > Hello authors, thanks for you reply. Given the modifications that are promised here I am happy to change my score to a full accept (7). Cheers,

---

### Official Review · Reviewer_UaTx · 2021-07-21

**Rating:** 7
**Confidence:** 2

**Summary:**

Disclaimer: I am not an expert in theoretical RL. My review is thus restricted to the overall soundness of the manuscript.

The authors describe theoretical work proposing a new sampling strategy for the online linear Q* problem making it sample efficient.
They modify and extend the LSVI-UCB algorithms for linear MDP and show via regret analysis, that their sampling scheme with state revisiting leads to sample efficient algorithms -  a feature missing in previous studies in this particular setting with classic episodic sampling.



**Main Review:**

The authors present an algorithm extending LSVI-UCB.
The LSVI-UCB algorithm exploits the assumption that both the reward, transition are linear functions of feature maps to derive closed form updates for the Q parameters via - quoting the paper -a ridge regularized least squares problem tailored to solving the Bellman optimality problem. Exploration bonuses are added to the resulting Q value that reflect the uncertainty in the estimation of Q.

The extensions are twofold
1) the ridge-regression type updates for the Q* parameters are done ‘bootstrapping’ from the bonus-less version Q* and not the upper bound.
2) a new episode sampling  protocol : having sampled k episodes, selected a particular h in [0,H-1], episode k+1 consists of episode k copied till h, then completed with H-h steps starting from the initial state in episode k at state h.

The manuscript is well written, background and previous work on which this work is built on are discussed and summarized. The contribution and its consequences are also clear and thorough.
I can’t comment much on the novelty or the details of the regret analysis and set my confidence score accordingly.

Some comments to (possibly) improve the manuscript
- the detailed algorithm 2 & 3 help a lot to understand the algorithm. Is there a typo in the update_remaining? Index j should be on the Qs
- it took me a while to understand when states are revisited, that it depended on the algorithm variables (bonus and gap), and how it happened (along the backward pass, this is hidden in alg2 and line 250). The presentation does not reflect that dependence and until the ‘our algorithm’ section this is not described. I’d suggest to extend on that and to change line 4 of Alg 1 to reflect that.
- This might be naive, but a broader and earlier discussion of the suboptimality gap (mentioned  from the start and defined page 40) would be helpful.

------------------------------------
## post rebuttal and discussion

I have read the response (thanks), the other reviews and the discussion and maintain my appreciation of the manuscript


**Time Spent Reviewing:**

5

---

> ### Author Response · Authors · 2021-08-10
> **Thanks for the positive evaluation of our paper as well as the valuable suggestions and pointing out the typo.**
>
> We will expand our discussion on (i) when states are revisited and (ii) how it happens, in order to enhance readability. We will also provide a broader and earlier discussion of the suboptimality gap to better set the stage for discussion.

---

### Decision · Program_Chairs · 2021-09-27

**Decision:**

Accept (Poster)

**Comment:**

The paper studies the RL problem under Q* linear realizability assumptions. This setting has been extensively studied in a stream of recent papers detailing impossibility results as well as positive results depending on whether access to a generative model and large suboptimality gaps are available or not. The authors make a non-trivial contribution in showing a positive (i.e., sublinear regret) in a novel online setting where the agent is allowed to to revisit previously traversed states. This setting is somewhat in between the full online and generative model scenarios and it is motivated in problems where the agent is not allowed to pick an arbitrary state in advance but it can record states and reset the system to states that have been visited in the past (e.g., games). While the scope of this setting could be debatable, it is overall reasonable and it sheds more light in the intricate limits of learnability in the Q* linear realizable case. Also, the theoretical and algorithmic contributions are non trivial. Overall, I believe the paper is likely to encourage further discussions in the community and inspire further research on the topic.

From rebuttal and the extensive discussion with reviewers, there are a number of elements that the authors should integrate in the camera ready version:
- The current main theorem is somehow difficult to parse as it "mixes" sample complexity and regret. I suggest to either explain these two contributions better or first summarize the overall result as a sublinear regret bound and then explain how it is obtain.
- As pointed out in the discussion, the final regret depends on 1/Delta^2. This suggests that the worst-case bound may be of order O(K^{2/3}). While the focus is actually to prove "learnability" (i.e., sublinear regret) and this may be overall unavoidable, I think it's worth pointing this out explicitly as it is a clear venue where people may focus to improve the current result.
- Acknowledge explicitly other limitations in the current version (e.g., knowledge of the gap). Again, this is not the first time this assumption is made, but it is worth making it very clear so as to encourage further research in this direction.
- Be sure to properly contrast your setting and results with Weisz et. al 2021a.
- The reviewers pointed out a few things that may need further clarification.